# *Saccharomyces cerevisiae* Mus81-Mms4 prevents accelerated senescence in telomerase-deficient cells

Erin K. Schwartz[1¤a], Shih-Hsun Hung[1], Damon Meyer[1¤b], Aurèle Piazza[1¤c], Kevin Yan[1], Becky Xu Hua Fu[1¤d], Wolf-Dietrich Heyer[1,2]*

**1** Department of Microbiology and Molecular Genetics, University of California, Davis, Davis, California, United States of America, **2** Department of Molecular and Cellular Biology, University of California, Davis, Davis, California, United States of America

¤a Current address: Santa Clara University, Department of Biology, 500 El Camino Real, Santa Clara, California, United States of America
¤b Current address: California Northstate University, College of Health Sciences, Sacramento, California, United States of America
¤c Current address: CR CNRS UMR5239, Team Genome Mechanics,Laboratory of Biology and Modelling of the Cell, Ecole Normale Supérieure de Lyon, allée d'Italie, FRANCE
¤d Current address: Department of Urology, University of California, San Francisco, San Francisco, California, United States of America
* wdheyer@ucdavis.edu

**Data Availability Statement:** All relevant data are within the manuscript and its Supporting Information files.

## Abstract

Alternative lengthening of telomeres (ALT) in human cells is a conserved process that is often activated in telomerase-deficient human cancers. This process exploits components of the recombination machinery to extend telomere ends, thus allowing for increased proliferative potential. Human MUS81 (Mus81 in *Saccharomyces cerevisiae*) is the catalytic sub-unit of structure-selective endonucleases involved in recombination and has been implicated in the ALT mechanism. However, it is unclear whether MUS81 activity at the telomere is specific to ALT cells or if it is required for more general aspects of telomere stability. In this study, we use *S. cerevisiae* to evaluate the contribution of the conserved Mus81-Mms4 endonuclease in telomerase-deficient yeast cells that maintain their telomeres by mechanisms akin to human ALT. Similar to human cells, we find that yeast Mus81 readily localizes to telomeres and its activity is important for viability after initial loss of telomerase. Interestingly, our analysis reveals that yeast Mus81 is not required for the survival of cells undergoing recombination-mediated telomere lengthening, *i.e.* for ALT itself. Rather we infer from genetic analysis that Mus81-Mms4 facilitates telomere replication during times of telomere instability. Furthermore, combining *mus81* mutants with mutants of a yeast telomere replication factor, Rrm3, reveals that the two proteins function in parallel to promote normal growth during times of telomere stress. Combined with previous reports, our data can be interpreted in a consistent model in which both yeast and human MUS81-dependent nucleases participate in the recovery of stalled replication forks within telomeric DNA. Furthermore, this process becomes crucial under conditions of additional replication stress, such as telomere replication in telomerase-deficient cells.

**Funding:** Work in the WDH laboratory was supported by grants GM58015 and CA92776 from the US National Institutes of Health. EKS was partially supported by a fellowship from the HHMI-IMBS training grant at UC Davis. DM was partially supported by fellowship 15IB-0109 from the CBCRP. SHH is partially supported by a fellowship from the Academia Sinica, Taiwan. The funders had no role in study design, data collection, and analysis, decision, to publish, or preparation of the manuscript.

**Competing interests:** The authors have declared that no competing interests exist.

## Author summary

Cancer cell divisions require active chromosome lengthening through extension of their highly repetitive ends, called telomeres. This process is accomplished through two main mechanisms: the activity of an RNA-protein complex, telomerase, or through a telomerase-independent process termed alternative lengthening of telomeres (ALT). Human MUS81, the catalytic subunit of a set of structure-selective endonucleases, was found to be essential in human cells undergoing ALT and proposed to be directly involved in telomere lengthening. Using telomerase-deficient *Saccharomyces cerevisiae* cells as a model for ALT, we tested the hypothesis that Mus81-Mms4, the budding yeast homolog of human MUS81-dependent nucleases, is essential for telomere lengthening as proposed for human cells. Using genetic and molecular assays we confirm that Mus81-Mms4 is involved in telomere metabolism in yeast. However, to our surprise, we find that Mus81-Mms4 is not directly involved in recombination-based mechanisms of telomere lengthening. Rather it appears that Mus81-Mms4 is involved in resolving replication stress at telomeres, which is augmented in cells undergoing telomere instability. This model is consistent with observations in mammalian cells and suggest that cells undergoing telomere shortening experience replication stress at telomeres.

## Introduction

The careful maintenance of telomere DNA is required for chromosome stability and to limit cell proliferative potential in eukaryotes [1]. For most cells, telomere shortening continues until reaching a critical point where the cells senesce, thus limiting cell division. However, in the case of stem cells or cancer cells–which have increased proliferative potential–continued cell growth is achieved through active telomere lengthening. Two mechanisms have been described to maintain telomere length including telomerase and alternative lengthening of telomeres (ALT) which utilizes homologous recombination to extend telomere ends [2, 3]. Understanding the unique processes by which recombination-mediated ALT extends telomere DNA is therefore critical to understanding ALT-dependent cancer cell growth.

Structure-selective endonucleases are key candidates for functioning in the process of ALT. Coordinated by the Fanconi anemia protein SLX4 (BTBD12 or FANCP), four distinct catalytic endonucleases (*MUS81*, *YEN1*, *RAD1*, and *SLX1*) are critical for processing of DNA joint molecules during DNA recombination, replication and repair (reviewed in [4]). Interestingly, several groups have reported the association of SLX4 with telomere DNA, which is dependent on the telomere binding protein TRF2 [5, 6]. Consistent with these results, human SLX4 has been shown to be an important factor for postreplicative DNA repair and mitotic fidelity in telomerase positive and ALT cells [7, 8]. In mammals, SLX4 physically interacts with a host of endonuclease complexes, including SLX1, MUS81, and XPF (Rad1 in budding yeast) [6, 9–13]. The interaction between SLX4, TRF2, and specifically MUS81 was found to be required for the telomere sister chromatid exchange observed in U2OS ALT cells [5, 14]. Co-localization of human SLX4 and associated nucleases, MUS81 and SLX1, with telomere DNA further supports a role for endonucleases in telomere stability [15].

MUS81 is a highly conserved endonuclease that cleaves DNA joint molecules (reviewed in [4, 16]). MUS81 activity requires the binding of an accessory protein, either EME1 or EME2 subunits in human cells [17], or Mms4 in budding yeast [18]. *In vivo* characterization of human and yeast MUS81-EME1/EME2 (Mms4) implicated the complex in processing of recombination intermediates at stalled replication forks and during meiotic crossover formation [19–22] The observed recombination-dependent lethality with defects in yeast DNA helicase, Sgs1,

points to a role of Mus81 in the processing of late recombination intermediates [23–34]. Physical interactions with budding yeast recombination protein Rad54 and fission yeast replication checkpoint kinase Cds1 (budding yeast Rad53, human CHK2) places Mus81 at the intersection of recombination and replication [19]. Specifically, recent studies have found an important role for human MUS81 in processing DNA substrates in regions undergoing mitotic DNA synthesis (MiDAS) that results from persistent replicative stress [35–37].

Human MUS81 has been reported to be a major contributor to ALT-mediated telomere recombination [14, 21]. Physical binding of MUS81 to telomeres, as well as colocalization of MUS81 with telomere-containing promyelocytic leukemia bodies in ALT cells provided evidence of a direct role for MUS81 in human ALT [14, 38]. However, other reports have found more general association of human MUS81 to telomere DNA in both ALT and non-ALT cells [5]. In addition, both human and *Arabidopsis thaliana* data suggest MUS81 restricts telomere length, contrary to what you would expect given a role in ALT [5, 39]. These results question the role for MUS81 exclusively in human ALT and open the possibility that MUS81 may have other functions in telomere maintenance.

Observations that budding yeast *MUS81* is not required for a process implicated in human ALT further confounds our understanding for how Mus81 may be functioning at telomeres. Break-induced replication (BIR) constitutes a sub-pathway of homologous recombination resembling recombination-dependent replication in bacteria and phage [40–43]. The process involves Rad51-dependent DNA strand invasion and DNA synthesis extending from the initial D-loop all the way to the telomere [44–46]. BIR has been specifically implicated as the underlying mechanism in ALT, possibly involving interactions between the telomere and extrachromosomal circular DNA containing telomeric repeats [45, 47, 48]. In budding yeast without functioning telomerase, there are two possible mechanisms by which cells can extend their telomeres; both depend on homologous recombination through a process related to BIR [45, 49–52]. Interestingly, genetic analysis of BIR demonstrated that Mus81 is not required for BIR in otherwise wild type budding yeast cells [29, 53]. This leaves open the question by which mechanism the MUS81 nuclease is involved in telomere maintenance in human ALT cells.

To directly assess the role for MUS81 in recombination-mediated telomere lengthening, we tested the contribution of *S. cerevisiae* Mus81-Mms4 to cell survival and growth in the absence of functional telomerase. Yeast provide an excellent model for ALT as they show similar recombination-mediated lengthening, as well as elevated levels of telomeric repeat-containing RNA (TERRA), characteristic of human ALT [54]. Similar to human cells, we find that yeast Mus81 readily localizes to telomeres in the absence of telomerase and is important for growth and viability after initial loss of telomerase activity. Further analysis revealed that *S. cerevisiae* Mus81, though potentially involved, is not an essential component of recombination-mediated telomere extension in yeast. Rather, *MUS81* becomes critically important for viability in the combined presence of telomere dysfunction and replicative stress. Genetic analysis with the replication factor, Rrm3, supports a model whereby Rrm3 and Mus81-Mms4 function in parallel pathways to support telomere replication. Combining these results with the reports in humans [14, 21], we propose a model, whereby both yeast and human Mus81 promotes replication fork stability and repair at telomeres, which becomes even more critical as telomeres undergo telomerase loss and telomere shortening.

## Results

### Catalytic activity of the Mus81-Mms4 structure-selective endonuclease contributes to cell growth and survival in the absence of functional telomerase

In budding yeast *S. cerevisiae*, loss or mutations of the genes encoding the RNA or protein components of telomerase (*TCL1* and *EST1-3*, respectively) results in progressive telomere

shortening, replicative senescence and in most cases cell death after approximately 50 generations [55–57]. However, a small fraction of cells recover and escape senescence by employing a recombination-mediated telomere lengthening process, termed Type I or Type II survivors in yeast [45, 49–52]. In human ALT cells, MUS81 has been shown to physically associate with telomeres in G2 and is required for telomere recombination and proliferative potential [14]. These results suggested a direct role for MUS81 in telomere recombination in the absence of telomerase. To assess the function of Mus81-Mms4 in the kinetics of replicative senescence in budding yeast, we combined *mus81Δ*, *mms4Δ*, or the catalytic-deficient *mus81-D414,415A* (*mus81-dd*, catalytic deficient allele of *MUS81* [58]) mutants with telomerase-negative mutants devoid of the catalytic subunit of budding yeast telomerase, *est2Δ*. We continued to performed serial liquid growth analysis over 80 generations.

Wild type cells propagated and grew to maximal titer overnight, approximately $1x10^8$ cells/ mL, showing little deviation over the 10-day period, or approximately 90 generations (Fig 1A). In an *est2Δ* mutant background, cells showed reduced proliferative potential within 50 generations, associated with an increased frequency of enlarged senescent cells (S1A Fig). Reduced proliferative potential of *est2Δ* mutant cells coincides with a progressive reduction in cell density, to a maximal 87% reduction compared to wild type at the lowest documented cell density (t(26) = 9.97, P<0.0001, Fig 1A). Recovery and survivor formation was observed as an increase in viability and growth, achieving almost wild type levels of cell density by generation 70 (Fig 1A).

Loss of *mus81Δ* in a telomerase-deficient background resulted in a 77% reduction in the minimal cell density compared to *est2Δ* mutants alone (t(43) = 3.09, P = 0.0035, Fig 1A and 1D). Similar reductions in cell density were observed for *est2Δ mms4Δ* and *est2Δ mus81-dd* double mutants, suggesting that the role for Mus81 requires heterodimer formation with Mms4 and catalytic activity of the complex (Fig 1B and 1D). The *est2Δ mus81Δ* double mutants showed similar morphological changes as the *est2Δ* single mutant (S1A Fig). Kinetics of recovery from senescence for the above single and double mutants appeared similar to *est2Δ* mutant cells (Fig 1A). Early onset and severity of replicative senescence was specific to a deficiency in the Mus81-Mms4 structure-selective endonuclease, as deletions of *YEN1* or *RAD1* did not result in the additive effect observed with *est2Δ* mutants (Fig 1C and 1D). Interestingly, there was a 73% reduction in cell density observed in *est2Δ slx1Δ* double mutants (t(43) = 2.91, P = 0.0058, Fig 1C and 1D), supporting an involvement of yeast Slx1-Slx4 in telomere stability under these conditions, similar to what has been observed in human cells [5–7, 59]. Double mutant analysis between *mus81-dd* and *slx1Δ* in the absence of telomerase resulted in an additional 59% reduction in cell density compared to *est2Δ mus81-dd* double mutant alone, suggesting independent roles in telomere stability (t(26) = 2.07, P = 0.049, S2A and S2B Fig).

## Mus81 function contributes to survival in telomerase-deficient yeast

During telomere shortening, the once clonal population becomes heterogeneous as each individual cell shortens their telomeres at slightly different rates. The relative reduction in the number of cell bodies can be affected by several variables including reduced cell division rates and/or cell survival. To discriminate between these possibilities, doubling time in generations was calculated from the colony counts and cell bodies were plated to ascertain viable colony forming units (S1B Fig). At maximal senescence (crisis), reduced proliferative potential of *est2Δ* mutant cells coincides with a progressive reduction in cell viability to a maximal 65% reduction compared to wild type (t(58) = 7.48, P<0.0001, Fig 2A; S1B Fig). Interestingly, the *est2Δ mus81Δ* double mutant exhibited an additional 56% reduction in cell viability compared to the *est2Δ* mutant alone (t(64) = 2.94, P = 0.0045, Fig 2B). This significant reduction in

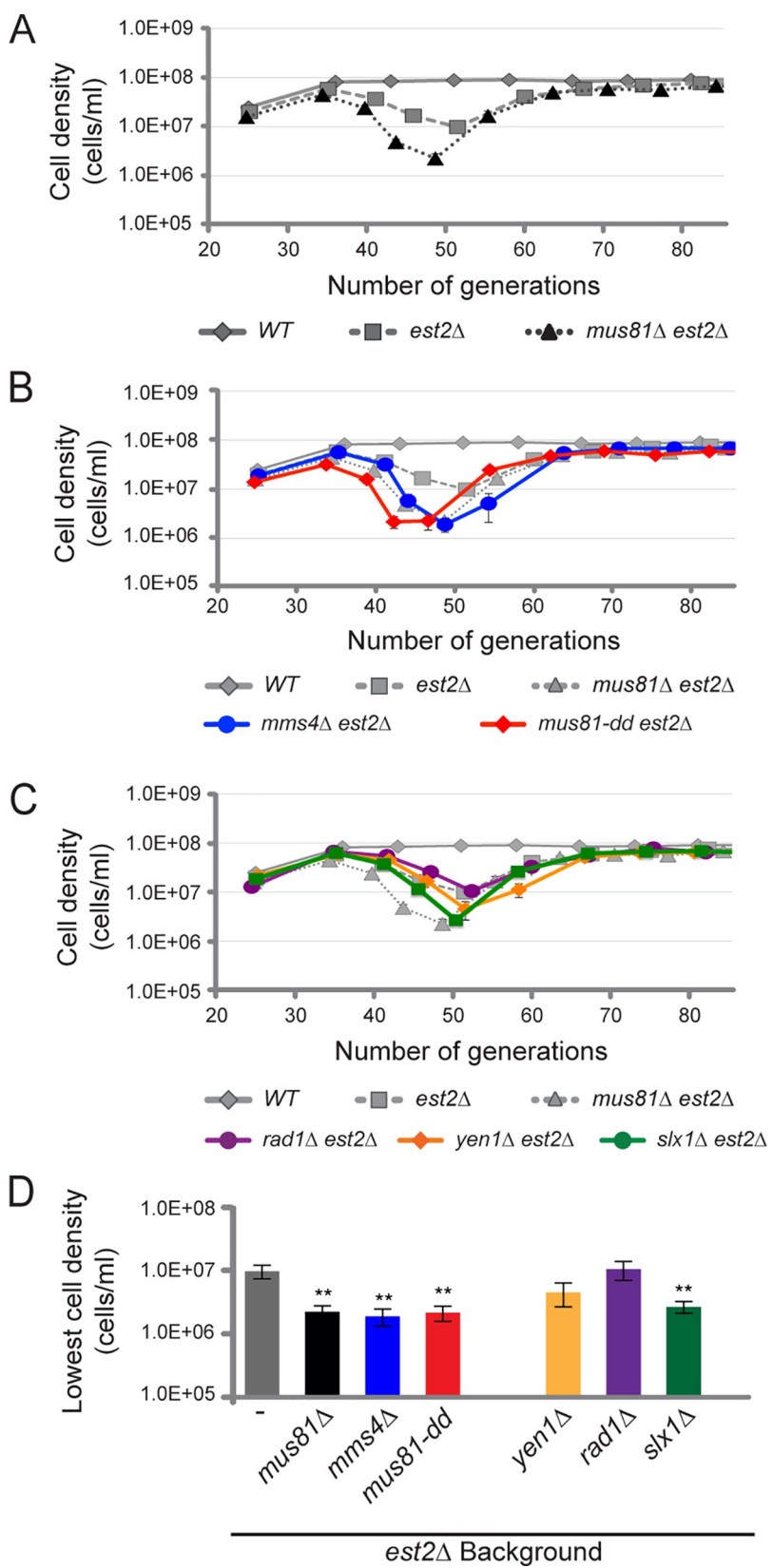

**Fig 1. The catalytic activity of Mus81 and Mms4 are required for cell viability and growth in the absence of telomerase.** Serial dilution assays monitoring cell density after 24 hours from an initial inoculate of $5 \times 10^5$ cells. For *A-C*, average cell density is plotted with one standard of error for (*A*) wild type (*WT*) (n = 20), *est2Δ* (n = 40), and *est2Δ mus81Δ* (n = 27), as well as (*B*) *est2Δ mms4Δ* (n = 8) and *est2Δ mus81-D414,415A* (n = 23). The Mus81 catalytic mutant, *mus81-D414,415A*, is represented by *mus81-dd*. (*C*) Additional analysis was completed on endonuclease mutants *est2Δ rad1Δ* (n = 7), *est2Δ yen1Δ* (n = 15), and *est2Δ slx1Δ* (n = 24). (*D*) The lowest average cell density was plotted for all strains analyzed in *A-C*. Statistics compare mean values to *est2Δ* single mutant. Haploid strains *est2Δ mus81Δ*, *est2Δ mms4Δ* were generated by sporulation of WDHY2961, as described in Materials and Methods. Similarly, *est2Δ mus81-dd* was derived from WDHY3007. Haploid strains *est2Δ slx1Δ*, *est2Δ yen1Δ*, and *est2Δ rad1Δ* were derived from WDHY3027, WDHY3036, and WDHY3145 diploids respectively. Additional *WT* and *est2Δ* haploids were derived from all five diploids. Student T-test is * P < 0.05, ** P < 0.01.

viability at senescence crisis was unique to Mus81, as it was not observed for any of the other endonucleases (Fig 2B).

Upon recovery, the viability of *est2Δ* is equivalent to wild type (t(11) = 1.51, P = 0.1592, Fig 2A). However, we observed a 14% reduction in viability of survivors with additional loss of *mus81* compared to *est2Δ* alone (t(8) = 5.75, P = 0.0004, Fig 2A). Surprisingly, there was no significant difference in doubling time with and without Mus81 across all generations, despite its known role in replication (S1C Fig). This together suggests that the reduced cell density observed in the absence of *MUS81* is largely due to reduced viability rather than an effect on cell division.

## Mus81 function in survival during telomere shortening is independent of Type I or II recombination-mediated lengthening

The reduced viability observed for mutant *mus81* in telomerase deficient yeast was intriguing and could have been attributed to a number of potential roles. One possibility is that the reduced viability observed in *est2Δ mus81Δ* mutants is due to Mus81's involvement in one of the survivor pathways in yeast. Human MUS81 has been proposed to be required for ALT and proliferation in telomerase deficient cells [14]. However, from our survival data, Mus81 does not appear to be essential for general recombination-dependent recovery (Fig 1B). This discrepancy may be explained by the existence of two alternative pathways in budding yeast that utilize recombination to extend their telomeres, designated Type I and II survivors, which differ in their molecular mechanism and genetic requirements (Fig 3A) [45, 52]. Surviving cells that use Type I lengthening have full-length telomeres that are proposed to originate from a single initiating and extension event in the subtelomeric Y'-elements. This process is similar to BIR and requires the activities of Rad52 and Rad51 (S3A Fig) [52]. Survivors using Type II recombination-mediated lengthening resemble human ALT, as it produces random telomere lengthening by amplification of the intra- or interchromosomal GC-rich telomere repeats involving also extrachromosomal circles containing telomeric DNA to facilitate telomere extension and cell proliferation (S3A Fig). Despite both survivor types being Rad52- or recombination-dependent they produce distinct telomere morphology that can be discriminated by Southern analysis or their genetic requirements [52].

To determine if Mus81 contributed to either Type I or Type II survivor formation, we evaluated the survival of *est2Δ mus81Δ* double mutants in the absence of *RAD51* or *RAD59*, essential genes for Type I or Type II survival formation respectively (Fig 3A; S3A Fig) [52]. In the absence of *rad51*, only Type II survivors form and if *MUS81* was required for Type II growth, the combined *est2Δ mus81Δ rad51Δ* triple mutant would be lethal as cells would be defective in both recombination-mediated survival pathways. Genetic analysis revealed the triple mutant recovered to the same cell density as the *est2Δ rad51Δ* double mutant, showing that Mus81 is not essential for Type II survivor formation (Fig 3B). Similar in concept to the

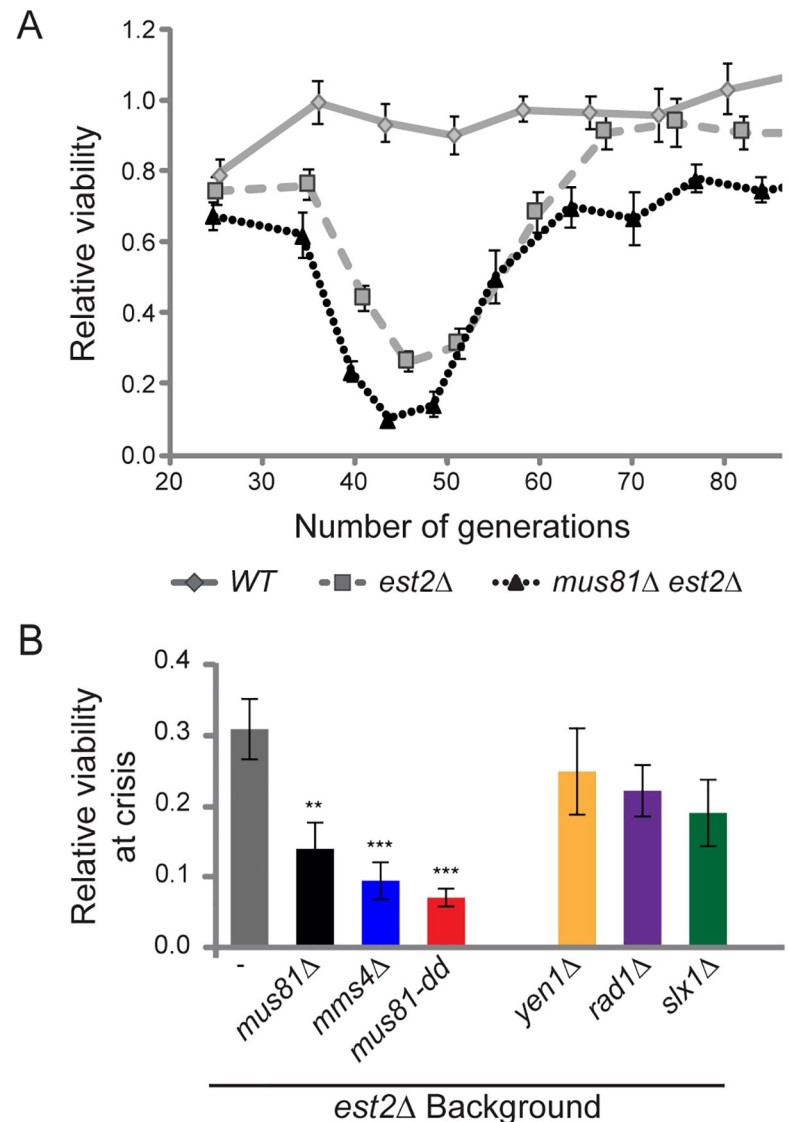

**Fig 2. Loss of Mus81 in the absence of telomerase results in reduced cell viability.** (*A*) Average relative viability with one standard error is presented for wild type (*WT*) (n = 20), *est2Δ* (n = 40), and *est2Δ mus81Δ* (n = 26). (*B*) Relative cell viability was calculated at the generation with the lowest average cell density (senescence crisis). The Mus81 catalytic mutant, *mus81-D414,415A*, is represented by *mus81-dd*. Statistics compare mean values to *est2Δ* single mutant. Haploid strains *est2Δ mus81Δ*, *est2Δ mms4Δ* were generated by sporulation of WDHY2961, as described in Materials and Methods. Using a similar method, *est2Δ mus81-dd* was derived from WDHY3007. Haploid strains *est2Δ slx1Δ*, *est2Δ yen1Δ*, and *est2Δ rad1Δ* were derived from WDHY3027, WDHY3036, and WDHY3145 diploids respectively. Additional *WT* and *est2Δ* haploids were derived from all five diploids. Student T-test is * P < 0.05, ** P < 0.01, *** P < 0.001.

previous genetic analysis, in an *est2Δ rad59Δ* double mutant background only Type I survivors will form and any requirement for *MUS81* in Type I survivor formation will become essential. Additionally, the double *est2Δ rad59Δ* and triple *est2Δ rad59Δ mus81Δ* mutant combinations both undergo senescence and recovery, suggesting Mus81 is not required in the formation of Type I survivors (Fig 3C). Statistical analysis of lowest cell density and its associated viability show no significant difference with and without *MUS81* during the formation of Type I or Type II survivors (S3B and S3C Fig). Slight deviations in the timing of cellular senescence and

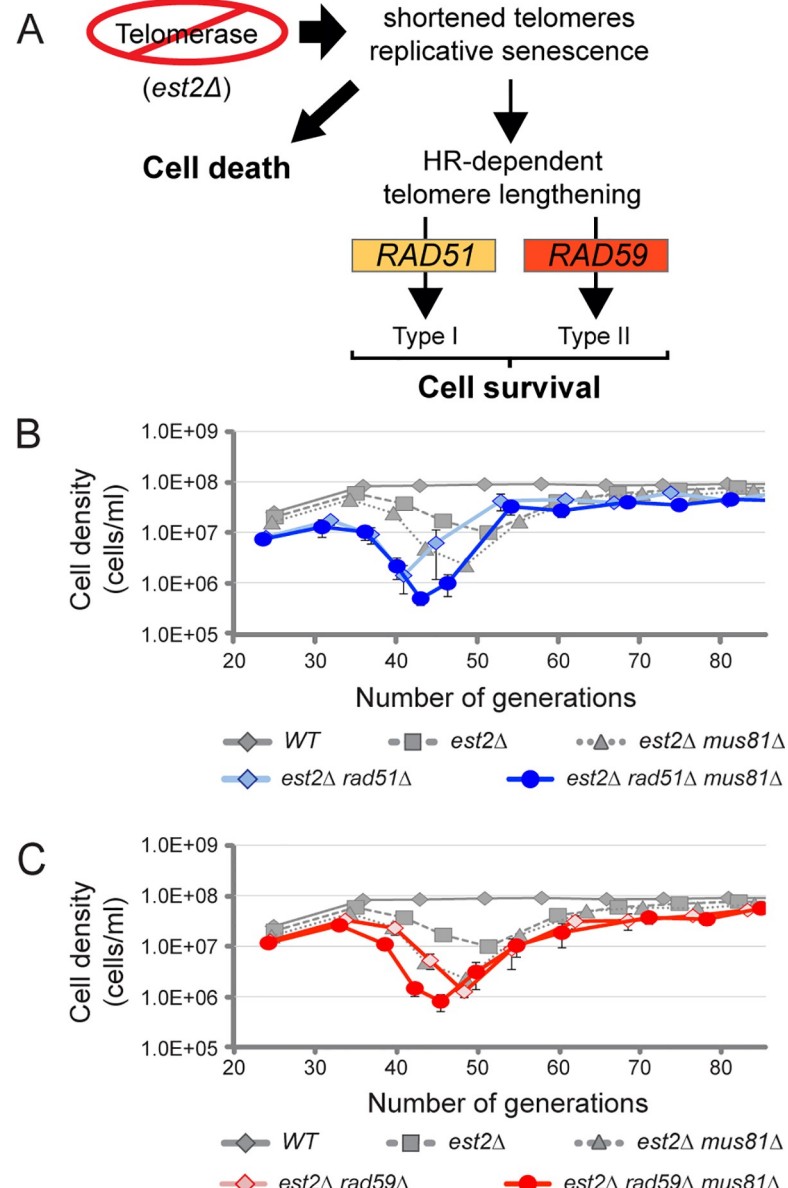

**Fig 3. Reduced viability in the absence of *MUS81* is independent of Type I or Type II telomerase-deficient survivor formation.** (*A*) Diagram of survivor formation in the absence of yeast telomerase. Genetic requirements are highlighted for *RAD51* and *RAD59* in Type I and Type II survivor formation, respectively. Arrow size represents abundance of cells entering cell death or survival *via* homologous recombination (HR)-mediated telomere lengthening. (*B*) Average cell densities are plotted with one standard of error for wild type (*WT*) (n = 20), *est2Δ* (n = 40), *est2Δ mus81Δ* (n = 27), and *est2Δ rad51Δ* (n = 8), and *est2Δ rad51Δ mus81Δ* (n = 10), or (*C*) *est2Δ rad59Δ* (n = 9) and *est2Δ rad59Δ mus81Δ* (n = 9). Yeast strains *rad51Δ*, *mus81Δ rad51Δ*, *est2Δ rad51Δ*, and *est2Δ rad51Δ mus81Δ* strains were derived from diploid WDHY3358 as described in Materials and Methods. Remaining haploid strains, *est2Δ rad59Δ*, *and est2Δ rad59Δ mus81Δ*, were derived from WDHY3366, and *est2Δ mus81-dd* was derived from WDHY3007. *WT* and *est2Δ* strains were derived from sporulation of all three diploids.

recovery were observed in the absence of *MUS81* and may suggest a partial role in these pathways.

To identify the telomere structure of survivors that form in the absence of *MUS81*, we evaluated the telomere morphology of surviving *est2Δ* single mutant and *est2Δ mus81Δ* double

mutant clones. We monitored the location of the subtelomeric Y'-element by Southern blot of *XhoI*-digested genomic DNA from at least 5 independent survivors. Due to the location of the *XhoI* restriction site and probe (diagramed in S3D Fig), we were able to visualize both subtelomeric elements (6.5 and 5.5 kbp) and telomeres (~1.3 kbp). Wild type cells had a distinct telomere length, which remained constant over 1 to 100 generations (S3E Fig). In the absence of telomerase, it has been observed that Type I survivors occur first, and then are soon outcompeted in liquid culture by Type II survivors resulting in a predominately Type II survivor population at later generations times (>80) [49, 50]. As expected, survivors in the *est2Δ* mutant strain exhibited random telomere lengths, characteristic of Type II formation (S3E Fig). Random telomere lengths in *est2Δ mus81Δ* double mutants were also characteristic of Type II survivors, independently demonstrating that *MUS81* was not essential for Type II survivor formation in budding yeast (S3E Fig).

## Mus81 endonuclease is involved in promoting cell growth in response to replication stress at telomeres

Our results show that Mus81 contributes to cell viability in the absence of telomerase; however, it remained unclear how Mus81 supports cell survival. Previous studies have found that Mus81 is required for replication fork restart and viability in response to agents that stall the progression of replication forks [19, 20, 22, 32, 60]. We hypothesized that Mus81 could promote accurate recovery of stalled forks at shortened telomeres that would otherwise have lethal consequences in subsequent cell divisions. To test this hypothesis, we first had to establish that there is increased genotoxic stress in the absence of telomerase.

Indeed, in the absence of yeast telomerase we were able to monitor phosphorylation of the DNA damage repair protein Rad53 (Fig 4A), indicative of DNA damage checkpoint activation [61]. Furthermore, loss of functional yeast telomerase resulted in increased genotoxin sensitivity to methyl methanesulfonate (MMS), but not hydroxyurea (HU) (Fig 4B). Additional loss of Mus81 in the absence of yeast telomerase resulted in a significant and reproducible viability defect in the presence of both replication stalling agents methyl methanesulfonate (MMS) and hydroxyurea (HU) (Fig 4B).

To test if this increased sensitivity to replication stalling agents extends to other situations of telomere disfunction, we evaluated if Mus81 was important for viability and growth in the situation of telomere uncapping. Cdc13 stabilizes telomere ends by recruiting telomerase, as well as coordinating leading and lagging strand DNA synthesis at the telomere [62–64]. Using a temperature sensitive hypomorphic allele of *CDC13*, *cdc13-1*, we assessed the ability of cells to survive under semi-permissive temperatures as single mutants and in combination with *mus81Δ*. Work from the Lundblad laboratory has carefully characterized the thermolabile activity of *cdc13-1* and shown that growth >34˚C results in additional defects in telomere function independent of Cdc13 [65]. Consequently, we conducted our studies at semi-permissive temperatures (<30˚C) well below this threshold, and any phenotypes observed are due to loss of Cdc13 function alone.

As expected, *cdc13-1* cells grown at permissive temperatures (24˚C) show comparable spotted cell density and colony size to wild type on nutrient-rich media (Fig 4C). Incubation at an intermediate, semi-permissive temperature (27˚C) produced a moderately defective Cdc13-1 protein, corresponding to a slight reduction in colony size on nutrient-rich media (Fig 4C). Single deletions of *mus81Δ* had similar spotted cell density and colony size as wild type at both permissive and semi-permissive under untreated conditions. Double mutants of *cdc13-1* with either *mus81Δ* or *mus81-dd*, exhibited cells with a significant and reproducible decrease in

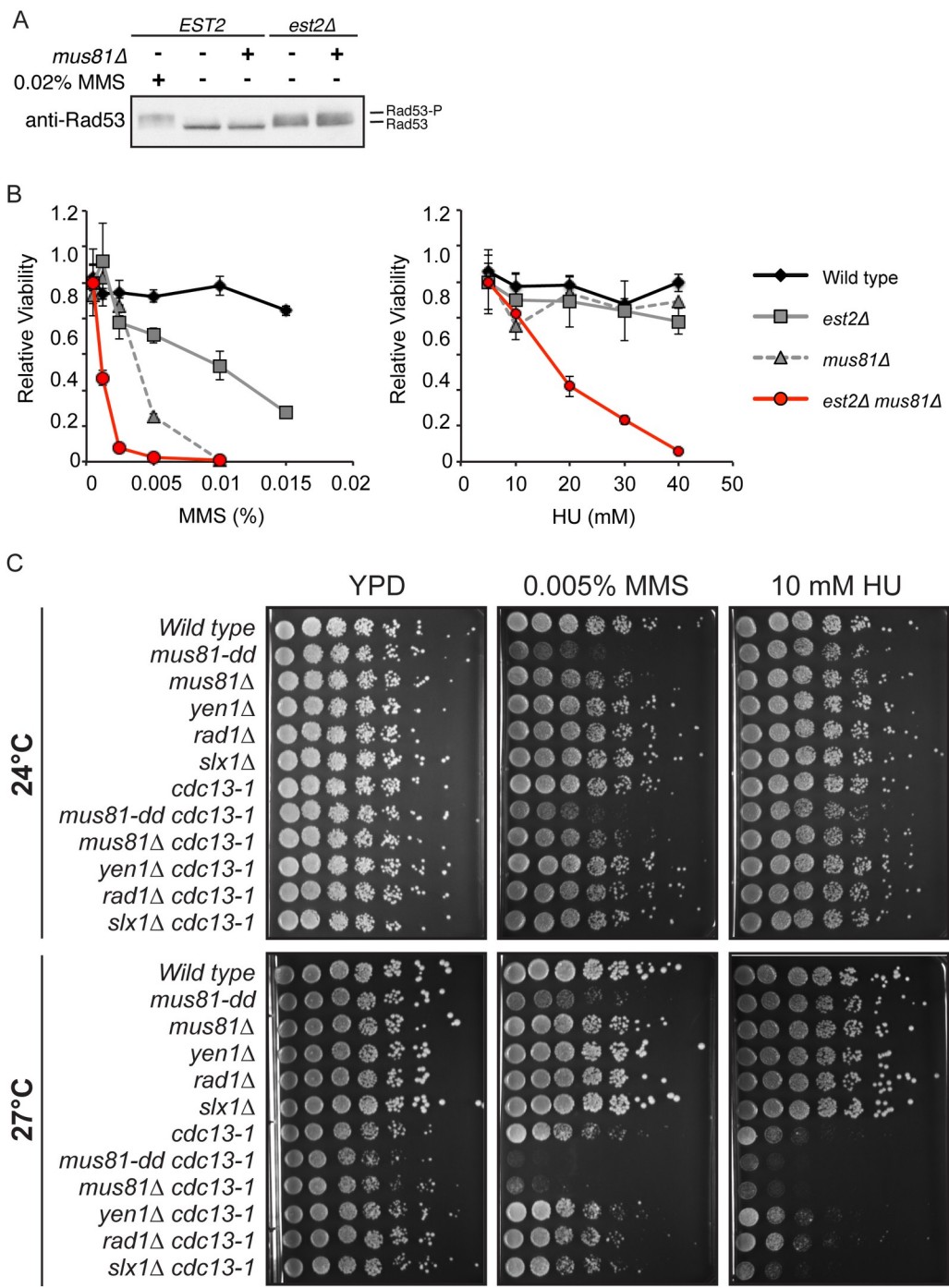

**Fig 4. Under conditions of telomere instability, *MUS81* is required for growth and viability in the presence of replication fork-stalling agents.** (*A*) Western blot using antibodies against the DNA damage repair kinase Rad53. Treatment of cells with 0.002% methyl methanesulfonate (MMS) was used as a positive control to activate the DNA damage response resulting in Rad53 phosphorylation as observed by a molecular weight shift. (*B*) Growth and viability of freshly derived haploid spores (~35 generations) was assessed using chronic exposure to increasing concentrations of methyl methanesulfonate (MMS) and hydroxyurea (HU). Average relative viability are plotted with one standard of error (n = 6). The strains were generated by sporulation of WDHY2961. (*C*) Growth and viability of various endonuclease mutants was assessed in the presence and absence of functional telomere capping protein Cdc13. Cells were chronically exposed to low doses of replication fork stalling agents, 0.005% methyl methanesulfonate (MMS), 10 mM hydroxyurea (HU) or no additions as control. Cells were incubated at permissive (24˚C) and semi-permissive (27˚C) temperatures for two to four days. Catalytic mutant, *mus81-D414,415A*, is represented as *mus81-dd*. Strains used are as follows: Wild type

(W303-*RAD5*, *MAT* α), *mus81-dd* (WDHY2835), *mus81Δ* (WDHY1858), *yen1Δ* (WDHY2755), *rad1Δ* (WDHY3106), *slx1Δ* (WDHY3148), *cdc13-1* (WDHY3086), *cdc13-1 mus81-dd* (WDHY3052), *cdc13-1 mus81Δ* (WDHY3058), *cdc13-1 yen1Δ* (WDHY3056), *cdc13-1 rad1Δ* (WDHY3054), and *cdc13-1 slx1Δ* (WDHY3112).

colony sizes compared to *cdc13-1* alone at the semi-permissive temperature in the absence of genotoxic treatment (Fig 4C).

In order to evaluate the possible role for Mus81 in telomere replication, we further challenged the yeast with low levels of replication fork stalling agents: methyl methanesulfonate (MMS) and hydroxyurea (HU). At the low genotoxin concentrations used, the single *mus81Δ* mutant exhibited little to no growth defect (Fig 4C), whereas single *cdc13-1* mutants showed a moderate growth defect at the semi-permissive temperature, suggesting that replication is indeed challenged in the absence of normal telomere capping (Fig 4C). Importantly, both double mutant combinations, *cdc13-1 mus81Δ* and *cdc13-1 mus81-dd*, resulted in severely reduced spotted cell density and colony size at the semi-permissive temperature in the MMS-treated condition and a moderate reduction in spot density the HU-treated condition (Fig 4C).

Genetic analysis was performed with a panel of deletion mutations for other known structure-selective endonucleases, including Yen1, Rad1, and Slx1. Despite the fact that these endonucleases have been shown to cleave similar substrates *in vitro*, the synergistic growth defects with *cdc13-1* appeared to be largely unique to Mus81 (Fig 4C). Consistent with the serial dilution data, loss of *YEN1* or *RAD1* had no effect on growth when combined with *cdc13-1*, in the presence or absence of replication stalling agents. However, there was an intermediate effect observed when combining *cdc13-1* with *slx1Δ* in HU that was not observed in MMS (Fig 4C). Combined, *MUS81* and to a lesser extent *SLX1*, appears to be promoting cell growth and/or survival during telomere instability, related to their role in replication fork support (Fig 1C; Fig 4C).

## Mus81 physically associates with telomeric DNA during the onset of replicative senescence

The synergistic growth defect observed between *mus81Δ* and both *est2Δ* and *cdc13-1* under conditions of replication stress suggests that Mus81 is functioning to support replication at shortening or uncapped telomeres. To test this idea we performed chromatin immunoprecipitation (ChIP) in *EST2* and *est2Δ* mutant cells with a MYC-tagged version of Mus81 (*MUS81-MYC*) or the catalytically inactive Mus81-dd (*mus81-dd-MYC*). As a negative control for Mus81 association at telomeres, we also performed the ChIP assay in corresponding untagged strains, *est2Δ MUS81* and *est2Δ mus81-dd*. Cells were serially grown as described above and harvested during log phase after approximately 45 generations, which represents the early phase of replicative senescence (Fig 1A). Fold enrichment of subtelomeric Y' element was calculated relative to the *SAM1* gene (Fig 5A). In the presence of functional telomerase, no detectable enrichment of telomere DNA was observed over background for either Mus81-Myc or mus81-dd-Myc (Fig 5B). However, upon loss of *EST2* Mus81-9Myc and mus81-D414,D415A-9Myc showed a 5-fold enrichment for telomeric DNA (Fig 5B). Taken together, these results suggest that Mus81 localizes to the telomere during replicative senescence.

As the role for Mus81 in telomere dysfunction appears to be linked to its role in replication, it follows that perhaps the requirement for human MUS81 is due to higher levels of replicative stress at telomeres in ALT cells. Given this hypothesis, we would predict an elevated level of Mus81 association in Type II survivors. Contrary to our prediction, we observe only background levels of enrichment at telomeres for both Mus81-9Myc and mus81-D414,D415A-9Myc in either Type II or Type I survivors (S4A and S4B Fig). However, we were able to detect

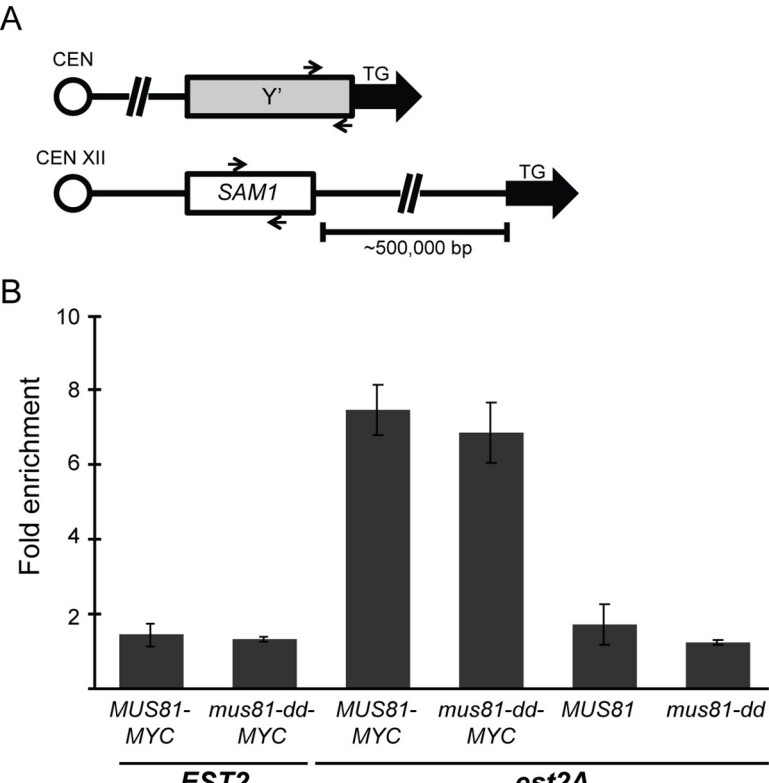

**Fig 5. Mus81 and Mus81-dd proteins associate with telomeric DNA in cells undergoing replicative senescence.** (*A*) DNA collected following ChIP was subjected to qPCR analysis using primers specific for *SAM1* (S) and the telomere proximal region of the subtelomeric Y' element (T). qPCR analysis of whole cell extract without the anti-Myc antibody using the S and T primers was used as a normalization control for each sample tested. Data represents fold enrichment as the normalized mean T/S ratio ± 1 SEM of at least two independent cultures for each genotype tested. (*B*) Association of Mus81 and Mus81-dd with telomeres was measured in *MUS81-9MYC* and *mus81-D414,D415A-9MYC* cells with either *EST2* wild type or *est2Δ*. Untagged versions (*est2Δ MUS81*, *est2Δ mus81-D414,D415A*) and *est2Δ mu81Δ* mutant cells were used as negative controls. Strains were derived from diploids WDHY2961, WDHY2962, and WDHY3007. Cells were collected during the early stages of replicative senescence and subjected to ChIP using anti-Myc antibody [108].

a reproducible and significant 2-fold increase in the telomere enrichment in telomerase-deficient survivors for the DNA damage response protein Esc4-9Myc (t(4) = 4.87, P = 0.008, S4C Fig). These results support an increased role for the DNA damage response pathway in survivors utilizing yeast ALT.

## Mus81 and Rrm3 function independently to stabilize replication forks in response to genotoxic stress

Rrm3 is a DNA helicase implicated in the support of replication through repetitive regions, including rDNA and telomeres [66]. In the absence of Rrm3, replication forks are more susceptible to stalling and pause more frequently in telomeric regions [66]. To evaluate if *MUS81* and *RRM3* function together to facilitate replication, we evaluated the growth and viability of cells lacking one or both genes under conditions of telomere shortening or uncapping.

Consistent with previous findings, loss of *RRM3* in the absence of telomerase had no significant difference for the lowest cell density observed during senescence (Fig 6A and 6B)[67]. However, loss of *RRM3* in the absence of telomerase and *MUS81*, resulted in a 79% reduction in cell density compared to the *est2Δ mus81Δ* double mutant (t(28) = 3.22, P = 0.003, Fig 6A

and 6B). Interestingly, this reduction in cell density was not due to a reduced viability at the lowest cell density (Fig 6C). Under conditions of telomere uncapping, loss of *RRM3* alone with *cdc13-1* at semi-permissive temperature showed a modest but reproducible reduction in spotted cell density and colony size which was greatly exacerbated by additional loss of *MUS81*, or the addition of replication stress (Fig 6D). For both conditions of telomere dysfunction, the contribution of *RRM3* to cell growth in the absence of functional telomeres is either more easily or only observed in the absence of *MUS81* (Fig 6A–6D).

To investigate a more general genetic interaction between that of *RRM3* and *MUS81*, we performed additional genetic analysis without telomere dysfunction. Tetrad analysis of individual haploid spores revealed that the concomitant loss of *RRM3* and *MUS81* resulted in slightly smaller spore clones, supporting a functional interaction between the two genes (S5A Fig). This phenotype is even more severe when using the catalytic mutant of Mus81 (S5A Fig). Chronic exposure to low concentration of replication fork stalling agents, MMS and HU, had little to no effect on the *mus81Δ* and *rrm3Δ* single mutants; however, the *mus81Δ rrm3Δ* double mutants exhibited moderately aggravated growth (S5B and S5C Fig). These results suggest Mus81 and Rrm3 are not epistatic, but rather function in independent or partially overlapping pathways to facilitate general replication fork progression and viability upon compromised replication fork progression.

More specifically, we addressed the interplay between Mus81 and Rrm3 during replication of the sub-telomeric and telomeric regions by performing two-dimensional gel electrophoresis (S5D Fig) [68]. As previously reported, the deletion of *RRM3* resulted in replication pausing at both the internal and distal telomeric TG-tracts (S5E and S5F Fig)[68]. Deletion of *MUS81* lead to a slight, although non-significant, decrease in replication pausing either alone or in combination with the *RRM3* deletion (S5E and S5F Fig). Hence, Mus81 does not promote progression of hindered replication forks at telomeres in the absence of Rrm3. This may be due to the particular nature of the replication fork block in the absence of Rrm3, such as tight protein-DNA roadblocks for the replicative helicase, different from the damages exerted genome-wide upon treatment with HU (stalled polymerases) or MMS (ssDNA nicks and abasic sites), which may not yield substrates for Mus81-Mms4. Combined these results suggest Mus81 and Rrm3 function in independent or partially overlapping pathways to facilitate replication fork progression and viability in the presence of replication fork barriers in situations of telomere dysfunction and more broadly.

## Discussion

In this study, we provide evidence for a function of *S. cerevisiae* Mus81 catalytic activity in promoting cell survival in the absence of functional telomerase (Fig 1A and 1D; Fig 2A and 2B). This was unexpected as previous attempts in yeast have not observe an effect of *MUS81* loss in the absence of the RNA component of yeast telomerase, *TLC1* [67, 69]. This is despite mutations in the telomerase *EST1-3* genes and *TLC1* RNA template showing identical telomere shortening phenotypes [55–57, 70]. The discrepancy between these studies may be explained by unknown molecular differences between the *est2Δ* and *tlc1Δ*. We explored this possibility by completing a serial dilution assay with *tlc1Δ* and *mus81Δ*. The senescence pattern of *tlc1Δ* and *est2Δ* were similar in severity and timing. Similarly, the accelerated senescence observed in the combined *tlc1Δ mus81Δ* double mutant matched the previous observations with *est2Δ mus81Δ* (S6 Fig and Fig 1).

An additional difference could have been the presence of a *rad5-535* mutation in the yeast background used in the previous studies [67, 69]. In yeast, the Rad5 DNA helicase ubiquitin ligase is involved in postreplication repair and has been shown to associate with telomere

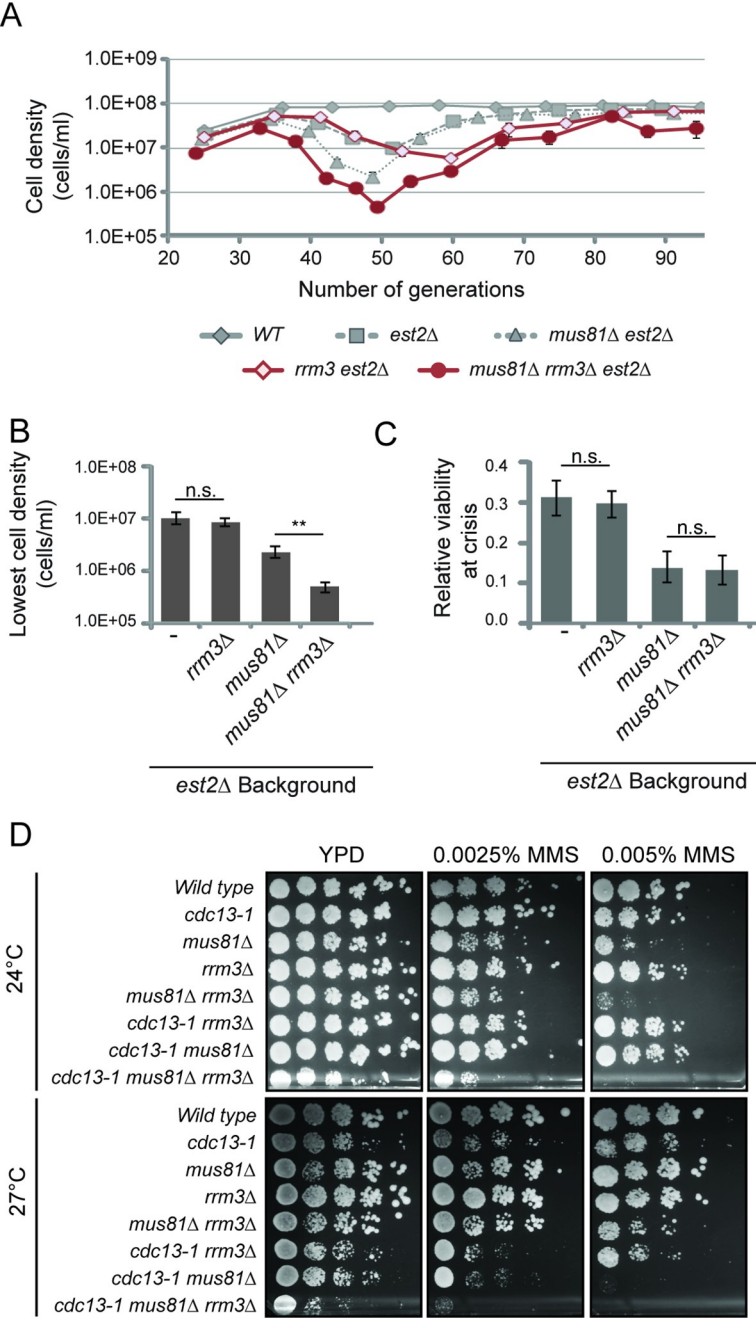

**Fig 6. Mus81 and Rrm3 act in separate pathways to support replication.** (*A*) Serial dilution assays monitoring cell density after 24 hours from an initial inoculate of 5x10⁵ cells. Average cell density and one standard error is plotted for wild type (*WT*) (n = 20), *est2Δ* (n = 40), *est2Δ mus81Δ* (n = 27), *est2Δ rrm3Δ* (n = 9) and *est2Δ mus81Δ rrm3Δ* (n = 9) strains. Many error bars are smaller than the plotting symbol. (*B*) The lowest average cell density was plotted for all strains in *A*. (*C*) Relative cell viability was calculated from colony forming units at the generation with the lowest average cell density for the strains in (*A*). Statistical comparison of mean values to *est2Δ* single mutant. Student T-test is * P < 0.05, ** P < 0.01. Haploid strains *est2Δ mus81Δ*, *est2Δ rrm3Δ* were generated by sporulation of WDHY3653 as described in Materials and Methods. (*D*) Growth and viability was assessed in the presence and absence of the telomere capping protein Cdc13. Cells were chronically exposed to low doses of methyl methanesulfonate (MMS) or no additions as control. Cells were incubated at permissive (24˚C) and semi-permissive (27˚C) temperatures for two to four days depending on growth. Strains used for *cdc13-1* genetics (C) are as follows: Wild type (W303-*RAD5*, *MAT* α), *cdc13-1* (WDHY3086), *mus81Δ* (WDHY2601), *rrm3Δ* (WDHY3638), *mus81Δ rrm3Δ* (WDHY), and *cdc13-1 mus81Δ* (WDHY3058). Strains WDHY2601 and WDHY3661 were mated to create fresh haploids for *cdc13-1 rrm3Δ* and *cdc13-1 mus81Δ rrm3Δ*.

DNA [69, 71]. However, the presence of *RAD5* or *rad5-535* did not appear to have an effect on the phenotypes observed in the senescence and recovery assay (S7A Fig). In fact, in the presence of telomerase combination mutations between *mus81* and *rad5-535* show reduced growth and viability in the presence of genotoxic stress, which is also observed for the *slx1* endonuclease (S7B Fig), suggesting endonucleases, *MUS81* and *SLX1*, as well as *RAD5* are generally required in the process of post-replicative repair. Additional experimentation in the *tlc1Δ mus81Δ* double mutant showed no difference between the *RAD5* and *rad5-535* containing genetics backgrounds (S6 Fig).

As the genetics of the telomerase mutants did not appear to explain the difference, it could be explained by differences in how the serial dilution assay was performed. This study used a serial dilution method with a starting cell density four times lower than analysis by Azam and colleagues which may have increased sensitivity to subtle changes in cell growth and viability [67]. Furthermore, whereas the previous study only counted cell bodies [67], we also collected viability data to identify a significant contribution for *MUS81* in regards to viability upon initial loss of telomerase (Fig 2A and 2B). Fallet and colleagues used different methods to monitor senescence and viability, and it is unclear if liquid growth or plating differences can account for the different results [69].

Following the report that human U2OS ALT cells required MUS81 for continued cell division [14], we predicted that Mus81 would be required for yeast Type II survivors which use a similar mechanism for telomere extension as human ALT. Interestingly, *S. cerevisiae* Mus81 was not essential for Type II, or Type I recombination-mediated telomere extension and survival (Fig 3B and 3C; S2A and S2B Fig). Changes in the timing of cell density crisis in *est2Δ mus81Δ rad51Δ* and *est2Δ mus81Δ rad59Δ* triple mutants (Fig 3B and 3C), suggests a partial involvement of *MUS81* telomerase-deficient survivor formation, but not an essential role as predicted. These results are consistent with previous epistasis analysis in the presence of telomerase demonstrating Rad51-dependent and Rad51-independent roles of Mus81 [72]. One way to interpret both roles is that Mus81-Mms4 processes Rad51-dependent joint molecules (Rad51-dependent function) and incises substrates (replication forks?) to initiate recombination (Rad51-independent function).

An ALT-independent role for MUS81 at telomeres has been proposed by other groups. In *Arabidopsis thaliana*, MUS81 was found to be involved in general telomere maintenance in combination with other nucleases [73]. Additional findings by Wan and colleagues proposed that human SLX4 and its associated endonucleases, including MUS81-EME1, were recruited by TRF2 to telomeres to regulate telomere length and fragility [5, 7, 59]. These results provided strong supporting evidence that structure-selective endonucleases, including MUS81, may be functioning at telomeres in processes independent of ALT. The implications of these findings in combination with the viability data suggest that the primary role for yeast Mus81 in promoting viability in telomerase deficient cells is by participating in processing DNA joint molecules during telomerase instability. Insight into this role was revealed genetically as Mus81 becomes critical for growth in yeast with telomere dysfunction (either loss of telomerase or uncapped telomeres) under conditions of replicative stress (Fig 4B and 4C). These key results suggest that Mus81, and possibly Slx1, support replication during times of telomere stress and presents an appealing model whereby Mus81 facilitates replication through regions that present obstacles to the replication machinery (Fig 7).

Telomeres have several unique structural qualities that make replication more difficult, including the multitude of telomere-specific proteins, repetitive DNA sequence and architecture of the DNA substrate [74, 75]. Because of these obstacles replication forks are observed to stall more frequently at telomeres in both yeast and humans, and often require homologous recombination as a key mechanism for the recovery of stalled or broken replication fork repair [68, 76–78] (reviewed by [79]). Under general replication stress, Mus81 is actively recruited to

stalled replication forks [19, 80] and is a key component of replication fork re-initiation in both yeast and mammalian cells [21, 60, 81]. It follows that Mus81 may contribute to replication fork stability at the telomere as it does at other locations within the cell.

Telomere dysfunction provides a unique opportunity to investigate the contribution of *MUS81* to conditions of increased replication stress. Telomerase loss leads to increased RNA-DNA hybrids and recombination at the telomere (reviewed in [82]). It follows that these added nucleotide joint molecules could act as barriers to the replication machinery. Consistent with this idea, loss of yeast replication supporting factor, *MRC1*, in a similar senescence assay as we performed here results in a more severe senescence crisis [83], similar to what we see with *mus81*. Likewise, loss of telomere capping protein (Cdc13) renders telomere ends susceptible to single-stranded DNA accumulation and recombination [84–87], again providing additional obstacles to replication.

Interestingly, both of these conditions depend on the catalytic activity of Mus81 for normal cell growth and viability (Fig 1A; Fig 4C). This reduced viability was aggravated when cells were further challenged with replication stress, even at replication stress levels where the *mus81Δ* single mutant strain had no growth defect alone (Fig 4B and 4C). Similarly, *MUS81* was critical in combination with *cdc13-1* mutation at semi-permissive temperatures, which was exacerbated in the presence of replication stress (Fig 4C). What Mus81 is exactly doing under these conditions is unclear. Although there is supporting evidence that Mus81 is acting on replication intermediates during replicative stress, it is also possible that the absence of Mus81 enhances the DNA damage checkpoint and that cells may not be able to progress through the cell cycle and adapt to reach the point where they can form survivors [88, 89].

As Rrm3 has been previously implicated in telomere replication, it was a logical progression to see if Mus81 was functioning in the same pathway or independently. Consistent with previous reports, loss of RRM3 in the absence of telomerase did not show any significant change in minimal cell density [67]. However, upon additional loss of *MUS81* we observed an aggravated growth defect, suggesting that Mus81 and Rrm3 function in independent and/or partially overlapping pathways to support telomeres during times of telomere shortening (Fig 6A and 6D). These results are consistent with a model where the helicase activity of Rrm3 migrates stalled replication forks, whereas Mus81-Mms4 would cleave replication-derived intermediates during replication fork re-initiation to support survival (Fig 7B). Defining the exact mechanism by which Rrm3 and Mus81 support telomere replication and replication of other highly repetitive regions is an intriguing area of future study.

It is interesting to speculate that the severe phenotypes associated with human MUS81 in ALT cells–which have aberrant recombination occurring at their telomere ends–is due to its role in replication re-initiation through the telomere. As a template for recombination, the telomere would be prone to the formation of DNA joint molecules, which may block replication fork progression [90] (Fig 7A). Indeed, ALT-associated PML bodies (APBs) readily colocalize with telomeres and proteins involved in replication stalling, including BRCA2, MUS81, FANCD2, and RAD51 [38]. The co-localization between APBs and repair proteins increases 2- to 3.5-fold in the presence of replication fork stalling agents, such as methyl methanesulfonate or hydroxy urea [38]. Specifically with regards to human MUS81-EME1, the endonuclease has been shown to co-localize and directly interact with WRN, a helicase required for replication through telomere DNA in ALT cells, and is critical for cell survival in the absence of *WRN* after events of replication fork stalling [91–93]. Interestingly, although we do observe an increase in Mus81 association at telomeres during telomere shortening, we did not observe a persistent increase in Mus81 association in Type I or Type II survivors (S4A and S4B Fig). This could be due to Mus81 not being required at telomeres in yeast cells performing ALT. Alternatively, it could also suggest a general increase in Mus81 association to other genetic regions, including our negative control region (*SAM1*) which would affect our enrichment.

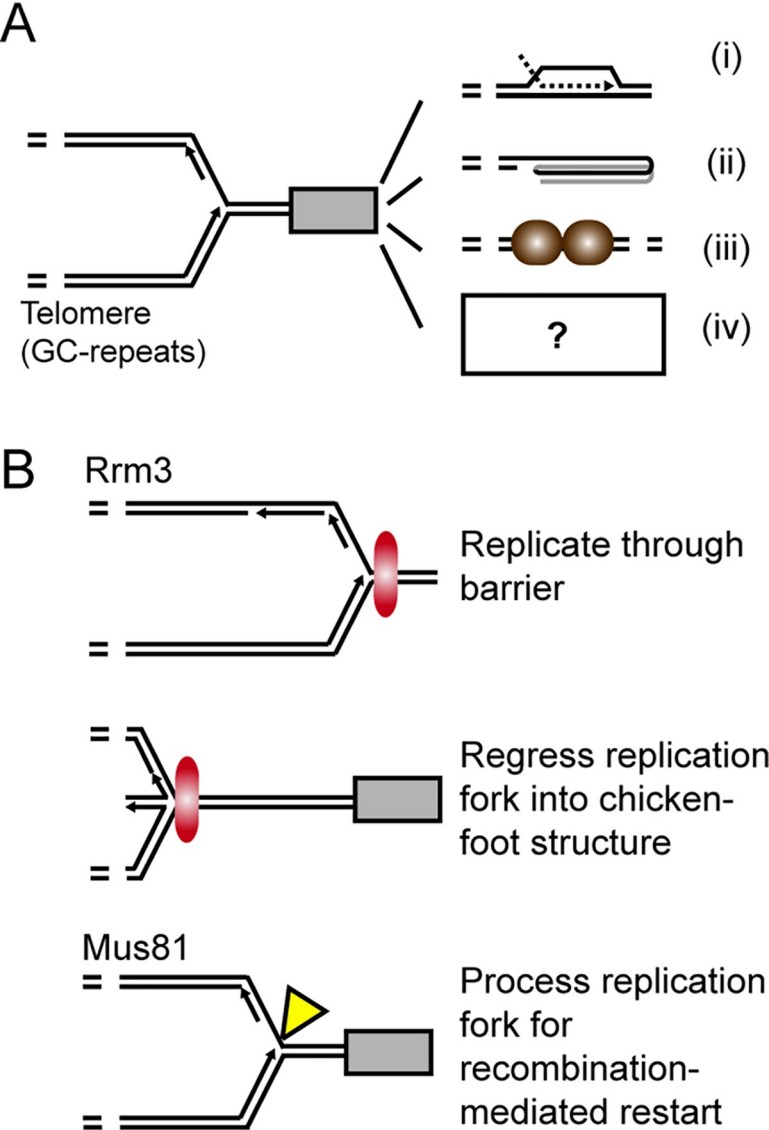

**Fig 7. Proposed model of Mus81 involvement at budding yeast telomeres.** (*A*) Telomere sequence and structure presents unique challenges to replication including (*i*) nucleic acid structures (*e.g.* t-loop, telomere RNA:DNA complexes), (*ii*) G-quadruplex structures, (*iii*) telomere-binding proteins and (*iv*) yet unknown sources of blockage. (*B*) Possible mechanisms to rescue stalled replication forks. Rrm3 is a 5'-to-3' DNA helicase required for normal replication fork progression through 'hard to replicate' sequences' including telomere and subtelomeric regions by directly removing barriers [66, 109]. A second mechanism envisions the regression of the stalled fork into chicken foot that is equivalent to a Holiday junction [110]. Thirdly, Mus81-Mms4 can cleave stalled forks and potentially lead to the formation of a substrate for recombination-mediated repair and restart [110].

Similar to human MUS81, *S. cerevisiae* Mus81 initially localizes to telomere DNA in the absence of functional telomerase (Fig 6B). In human cells, MUS81 accumulation at the telomere is dependent on the SLX4 endonuclease scaffold and telomere binding protein TRF2 [5, 6, 11]. An interaction of Mus81-Mms4 with the Dpb11-Slx4 complex was reported, but this interaction is counteracted upon activation of the DNA damage checkpoint [94]. We were able to observe a role for Slx1 in cell growth and viability in the absence of functional telomerase, which was independent of *MUS81* (Fig 1C; Fig 4C; S2 Fig). The involvement of Mus81 and Slx1 at the telomere was specific to these two structure-selective endonucleases and was not observed for either

Yen1 or Rad1 –endonucleases that have been shown to cleave similar DNA structures *in vitro* and have overlapping roles in processing meiotic DNA structures. However, we cannot exclude the possibility that Yen1 or Rad1 can partially compensate for the loss of Mus81 as has been seen in meiotic recombination [31]. Additional genetic analysis will need to be performed to evaluate overlapping roles for endonucleases in replication fork restart and telomere stability.

The specific involvement of Mus81 and Slx1 endonucleases at the telomere is intriguing as previous studies have shown a role for both proteins in replication of DNA regions prone to postreplicative repair. *S. cerevisiae* Mus81 and *Schizosaccharomyces pombe* Slx1, have previously been implicated in stabilizing replication through the highly repetitive ribosomal DNA (rDNA) [95, 96]. Loss of Mus81 results in three observations at rDNA: first, replication forks in rDNA stall more readily in response to protein-bound obstacles; second, cells accumulate recombination-dependent DNA structures at rDNA; and third, it results in rDNA repeat expansion [95]. One could imagine that these events at the telomere–which is also highly repetitive–would also have detrimental effects on telomere replication, especially if the telomere was being used as a template for telomere-end extension. Many replication barriers exist at the telomere that need to be overcome, including telomere-binding proteins, the protective T-loop structure, unique secondary structures that result from the repetitive, and GC-rich telomere sequence, as well as recombination-mediated structures involved in telomere lengthening (human ALT). It is interesting to speculate that Mus81 and Slx1 may participate together to overcome some of these obstacles in replication fork support at the telomere, through a yet undefined mechanism.

Furthermore, a role for yeast Mus81-Mms4 in supporting replication through telomere DNA complements a similar role proposed for human MUS81 at common fragile sites [37, 97–100]. Common fragile sites are associated with chromosomal regions that are particularly difficult to replicate and prone to the formation of gaps and breaks during metaphase [101–103]. Human MUS81 has been shown to recruit to these sites and is required for their proper segregation in U2OS, HeLa, and MRC-5 cells [97, 98]. MUS81 knockdown under conditions of replication stress leads to instability of these sites, observed by increased numbers of DNA double strand breaks and chromosome mis-segregation in subsequent G1 [37, 97, 98]. Specifically, human MUS81-EME1 has been found to physically and functionally interact with the RECQ5 helicase to promote mitotic DNA synthesis, termed MiDAS, that results from persistent replicative stress [35–37]. Given the role for both yeast Mus81 and Slx1 in telomere and rDNA replication, it will be interesting to evaluate the possibility of a parallel role for human SLX1-SLX4 in processing common fragile sites, and how these endonuclease complexes function together to support replication in these hard-to-replicate regions [99].

In summary, we present evidence that yeast Mus81-Mms4 contributes to cell viability in the absence of functional telomerase. We further show that this involvement is not through recombination-mediated telomere lengthening as proposed in humans [14, 104], but rather through an indirect role involving replication stress. These findings can be combined with previous results to provide a consistent model where human and yeast Mus81-Mms4/EME1 participate in telomere replication as it encounters unique protein and unusual nucleic acid structures in situations of telomere instability. These and other results suggest that Mus81 localization to the telomere may be a promising indicator for telomere dysfunction.

## Materials and methods

### *S. cerevisiae* strains and plasmids

A list of all strains used for this study is presented in S1 Table. Strains were created using standard protocols for yeast growth and genetic manipulation. Genomic DNA from 1922-1B was used as the template to produce the *rad1::LEU2* cassette transformed into W303-*RAD5 MAT*α

and WDHY3006 to create WDHY3106 and WDHY3145, respectively. WDHY3085 and WDHY3086 were spore clones from mating W303-*RAD5 MATα* and y816. *RAD5* status was confirmed by PCR. WDHY3639, WDHY3606 and WDHY3634 were spore clones from mating W6241-2A with W303-*RAD5 MATα*, and WDHY1858 with WDHY2835, respectively. Special considerations were made for both telomerase-deficient *est2Δ* and telomere compromised *cdc13-1* strains. As yeast telomerase is required for normal cell growth and viability, *est2Δ* is maintained as a *EST2/est2::ura3::LEU2* heterozygous diploid and sporulated to create the desired haploid mutant strains. These strains were used immediately for analysis or subsequent matings. All temperature sensitive *cdc13-1* strains and matings were maintained at room temperature (22˚C).

## Serial liquid growth and viability assay

Serial liquid growth assays were performed essentially as previously described [105]. Briefly, heterozygous *EST2/est2::ura3::LEU2* diploids were sporulated to create the desired haploid mutant strains. After two days, a small number of cells were transferred onto selectable media to determine genotype, while minimizing the loss of cells from each spore colony. Spore colonies with the desired markers were resuspended in 1 mL of sterile water and total cell densities determined by cell counting. The initial number of generations of the original spore colony was calculated from this value. Five milliliters of liquid YPD was inoculated with $5x10^5$ cells and incubated at 30˚C overnight with rotation. To determine cell viability, 100 or 500 cells were plated for wild type and *est2::ura3::LEU2*-derived strains, respectively. Colonies were grown for at least two days and up to five days, and visible colonies were counted. Cells undergoing senescence undergo variable cell growth forming colonies barely visible after two days, resulting in longer incubation periods to ease counting. Analysis of colony counts between two and five days revealed no significant differences in the total number of colonies (S8 Fig). The number of colonies divided by the original number of plated cells defined percent viability. Cell density multiplied by the percent viability defined the number of viable cell bodies. After 24 hours in liquid culture total cell bodies were counted and this value used to inoculate the next overnight culture and plating for cell viability. This process was repeated for 10 consecutive days, or until the calculated number of generations exceeded 80. Cell morphology was assessed using the Axioplan 2 imaging system and images analyzed with Improvision Volocity software (Perkin Elmer). All plots show average values of 7–48 independent spore clones for each genotype.

## Southern blot analysis

DNA extraction was performed on cell pellets collected from serial liquid growth cultures at various stages of senescence and therefore required 1–5 mL of overnight culture depending on the genotype. Five independent clones were selected after tetrad dissection and independently processed for serial dilution analysis. At approximately 25 and 80 generations, cells were pelleted and DNA extracted for Southern blot. Purified DNA was digested by *Xho*I. Following digestion, 10 μg of DNA was separated overnight on a 0.8% agarose gel with a 0.5% TBE buffer system overnight. The DNA was transferred onto nitrocellulose using capillary action and blotted with probes to the Y'-element (Fig 3B). The probes for Southern analysis were prepared using PCR with primers T3 (5'-AGC GCG CAA TTA ACC CTC ACT AAA G-3') and T7 (5'-CGT AAT ACG ACT CAC TAT AGG G-3') with pRS313/Y'*Rsa*I template, kindly provided by Dr. Gottschling [106]. PCR products were radiolabeled with $[\alpha-^{32}P]$ dCTP using the Random Primer DNA Labeling Kit (Roche). Radiolabeled signal was captured and visualized with the Storm860 PhosphorImager (Molecular Dynamics). Images were processed using ImageQuant TL software, v. 5.2 (Amersham).

## Chronic Genotoxicity Assays

Chronic genotoxicity exposure assays were preformed essentially as previously described [107]. Briefly, overnight cell suspensions were diluted to an $OD_{600}$ reading of 1.0 and plated as serial, five-fold dilutions onto nutrient-rich YPD media. Growth and viability effects of genotoxic exposure were performed in the presence of low-level methyl methanesulfonate (MMS) and hydroxyurea (HU). Telomere compromised *cdc13-1* strains were plated and grown at permissive (24˚C) and semi-permissive (27˚C) temperatures. All other plates were grown at 30˚C and images taken daily to monitor growth and viability.

## Chromatin Immunoprecipitation Assay (ChIP)

Newly dissected mutant spores of *est2::ura3::LEU2 mus81::KAN-MX*, *est2::ura3::LEU2 MUS81-9MYC* and *est2::ura3::LEU2 mus81-D414,D415A-9MYC* were subject to the serial liquid growth assay described above. Following growth for ~40 generations, cells from each culture were diluted to an $OD_{600}$ of 0.2 in 100 ml of fresh YPD and allowed to grow to an $OD_{600}$ of 1.0 at which time 50 ml of each culture was removed, pelleted, and immediately subject to the ChIP procedure described previously [108]. Briefly, the pelleted cells were exposed to formaldehyde, washed, lysed in a buffer containing protease inhibitors, and the DNA sheared to ~500 bp by sonication. The whole cell extract was subjected to immunoprecipitation by incubating with 5 μg anti c-MYC antibody (Santa Cruz Biotechnology) at 4˚C overnight with shaking. The next day, protein G agarose beads were added to the samples and incubated at 4ºC for three hours. The beads were then washed successively in SDS (50 mM HEPES pH 7.5, 140 mM NaCl, 1 mM EDTA pH 8.0, 0.025% SDS), a high salt buffer (50 mM HEPES pH 7.5, 1 M NaCl, 1 mM EDTA pH 8.0), and LiCl solution (20 mM Tris-Cl pH 7.5, 250 mM LiCl, 1 mM EDTA pH 8.0), and finally TE buffer (20 mM Tris-Cl pH 7.5, 0.1 mM EDTA pH 8.0). DNA/protein/antibody complexes were eluted from the beads with elution buffer (1% SDS/TE) and the DNA/protein cross-links reversed by incubation at 65ºC overnight. Proteins were then digested with proteinase K for two hours, extracted with phenol:chloroform, and the DNA precipitated with ethanol. Selected DNA sequences were analyzed by qPCR in triplicate using a Roche LightCycler480 with the following primer sets: a 198 bp *SAM1* sequence was amplified with the primers olWDH1653 (SAM1-5') (5'-CGA AGC TAA CCG AAA AAC AAC G-3') and olWDH1654 (SAM1-3') (5'-GCC CTT GCC TAC TAG TGC ATT T-3'), and a 195 bp sequence specific to the telomere proximal Y' element was amplified with the primers olWDH1655 (TEL-5') (5'-TAG AGA TTC GAG CAG AGA AG-3') and olWDH1656 (TEL-3') (5'-CCT ACT CTT TCC CAC TTG TC-3'). PCR products were quantified using the Roche LightCycler480 software. The *TEL* and *SAM1* PCR signals obtained by ChIP with strains containing the *MUS81-9MYC* and *mus81-D414,D415A-9MYC* alleles were normalized to *TEL* and *SAM1* PCR signals obtained from the above ChIP strains without antibody or input. The *mus81Δ* strain was used as a control and was subject to the same ChIP protocol and analysis as the MYC tagged versions of Mus81. The normalized *TEL* and *SAM1* PCR signals for each genotype tested were used to obtain the T/S ratios. The mean T/S ratios ± 1 standard error mean reported was derived from two or three independent clones for each genotype and repeated a minimum of four times. The *SAM1* PCR was chosen as the control locus for the *TEL* PCR because it lies distal to the telomere.

## Telomeric replication intermediates analysis by bi-dimensional gel electrophoresis

The strains used for telomeric replication intermediates analysis by bi-dimensional gel electrophoresis were W303-RAD5 (WT), WDHY5102 (*mus81Δ*), WDHY3638 (*rrm3Δ*), and

WDHY3605 (*mus81Δ rrm3Δ*). From exponential cultures in liquid YPD, $5 \times 10^8$ cells were harvested by centrifugation at 4°C, washed once in 10 mL of ice cold water, and frozen. Crosslinking of replication intermediates with Trioxsalen (Sigma-Aldrich, T6137) was performed as follow: each pellet was thawed in ice, re-suspended in 5 mL of a freshly made Trioxsalen solution (50 mM Tris-HCl pH 8.0, 50 mM EDTA pH 8.0, 20% ethanol 200 proof, 0.1 mg/mL Trioxsalen; from a 0.5 mg/mL stock solution in pure ethanol left in agitation overnight at room temperature for optimal solubilization), and split in two 60 mm petri dishes. Petri dishes were placed horizontally in ice 1–2 cm under a 365 nm UV lamp (Blak-Ray, B100A) and irradiated for 15 minutes with swirling every 5 minutes. Cells were transferred back in a falcon tube and each petri dish was washed with 2.5 mL of TE solution (50 mM Tris-HCl pH 8.0, 50 mM EDTA pH 8.0). Cells were pelleted at 4°C and frozen or immediately used for DNA extraction. Cells were resuspended in 1 mL of spheroplasting buffer (0.4 M Sorbitol, 0.4 M KCl, 0.5 mM MgCl$_2$, 40 mM Sodium Phosphate buffer pH 7.2) and spheroplasted upon addition of 5 uL of Zymolyase 100T solution (2% Glucose, 50 mM Tris-HCL pH 7.5, 5 mg/mL Zymolyase 100T (US biological, Z1004)) and incubation at 37°C for 30 minutes. Cells were pelleted at 8,000 rpm at room temperature, resuspended in 0.5 mL of lysis solution (20 mM Tris-HCl pH 8.0, 50 mM EDTA pH 8.0, 1% SDS, 0.2 mg/mL Proteinase K) and incubated 30 minutes at 65°C. After addition of 0.2 mL of ice-cold 5 M KAc, the lysate was incubated in ice for 1 hour and centrifuged at 16,000 g at 4°C for 10 minutes. The supernatant was recovered and nucleic acids precipitated with isopropanol. Dried pellets were incubated overnight at 4°C in 50 mL of TE pH 8.0 with 25 µg of RNAse A. Given its thickness, the pellets could not be fully resuspended. Consequently, the entire preparations were digested with 100 U of EcoRI-HF (R3101) and RNA eliminated upon incubation at 37°C for at least 4 hours, allowing resuspension. Approximately 30 µg was then migrated in a 0.4% low EEO agarose gel (Saekem Gold; poured at 65°C and solidified 30 minutes at 4°C) in TBE 1X at 45 V for 18 hours at room temperature. The gel was stained in an ethidium bromide bath for 30 minutes with shacking, and thin gel slices containing DNA fragments ranging from ≈5 to 20 kb were isolated for the second dimension. The second migration was performed in 0.9% agarose (Affymetrix, LE) 1X TBE with 0.3 µg/mL ethidium bromide at 250 V for ≈3h30 at 4°C. Psoralen crosslinking was reversed by irradiating the gel with UV 254 nm in a Spectrolinker (XL1500) for 10 minutes. DNA was transferred and crosslinked onto a Hybond-XL membrane (GE healthcare) according to the manufacturer instructions (alkali procedure and UV crosslinking). The membrane was blocked for 3 hours at 65°C in 40 ml of Perfect Hyb Plus hybridization buffer (Sigma-Aldrich, H7033). A 1,345 bp PCR fragment corresponding to the Y' element immediately proximal to the EcoRI site (amplified from genomic DNA using primers olWDH1782 (5'-GGG ACT TGC ATC AGT TGC-3') and olWDH1783 (5'- CTG TCACTG CTA TTG CTC TCC-3')) was labeled with [α-$^{32}$P]-dCTP (Perkin-Elmer, BLU-513Z) by random priming with the Decaprime II kit (Ambion) according to the manufacturer instructions. The denatured probe, together with 200 µg of cold carrier salmon sperm DNA, was added to the membrane in 30 mL of pre-warmed Perfect Hyb Plus buffer and incubated overnight at 65°C. The membrane was washed twice with pre-warmed washing buffer (40 mM Na$_2$HPO4 pH 7.3, 1% SDS, 2 mM EDTA) at 65°C for 10 minutes, and twice at room temperature for 15 minutes with washing buffer pre-warmed at 42°C. A phosphor-screen was exposed to the membrane for at least 24 hours, and scanned with the Storm PhosphorImager (Molecular Dynamics). Images were analyzed using ImageQuant TL.

## Supporting information

**S1 Fig. Cellular characteristics following loss of telomerase.** (*A*) Cell morphology was documented using bright-field light microscopy. Images were taken during early-senescence (~25

generations), late-senescence (~50 generations) and post-senescence (>80 generations). (*B*) Cell viability was monitored daily by plating 100 or 500 cells and allowing 2–5 days of growth for colony formation. Images were taken for visual documentation of colony growth at generation times 25 and 50. Post-senescence generation times were 80 or 100 for *est2*-deficient and wild type strains, respectively. (*C*) The number of cell divisions per 24-hour period are plotted over the total number of generations for wild type (*WT*), telomerase deficient *est2Δ* cells, and *est2Δ mus81Δ* double mutant strains. Error bars represent standard deviation of the number of cell divisions per 24-hour period for at least seven independent spore clones at each generation. All haploid strains were derived from WDHY2961 as described in Materials and Methods.
(TIF)

**S2 Fig. Mus81 and Slx1 contribute independently or in partially overlapping pathways to maintain cell growth in the absence of telomerase.** (*A*) Serial dilution assays monitoring cell density after 24 hours from an initial inoculate of $5\times10^5$ cells. Average cell density and one standard error is plotted at the given generations for wild type (WT) (n = 20), *est2Δ* (n = 40), *est2Δ mus81Δ* (n = 27), *est2Δ slx1Δ* (n = 24) and *est2Δ mus81-D414,415A (mus81-dd)* (n = 23) and *est2Δ mus81-D414,415A (mus81-dd) slx1Δ* (n = 7) strains. Haploid strains in (*A*) were generated by sporulation of WDHY3007 (WT, *est2Δ* and *est2Δ mus81Δ)* and WDHY3114 (*est2Δ slx1Δ* and *est2Δ mus81-D414,415A (mus81-dd))* as described in Materials and Methods. (*B*) Average cell density was plotted with one standard error at the generation with the lowest average cell density for strains in (*A*). Statistics compare mean values. Student T-test is \* P < 0.05. (TIF)

**S3 Fig. Mus81 is not required for Type I or Type II survivor formation.** (*A*) Proposed mechanisms for Type I and Type II alternative-telomere lengthening and involved proteins. Differences in survivor telomere length are highlighted at the bottom of the diagram. (*B*) The lowest average cell density and one standard error was plotted from serial dilution experiments shown in Fig 3 for strains *est2Δ* (n = 40), *est2Δ mus81Δ* (n = 27), *est2Δ rad51Δ* (n = 8) and *est2Δ mus81Δ rad51Δ* (n = 10), *est2Δ rad59Δ* (n = 9) and *est2Δ mus81Δ rad59Δ* (n = 9). Statistics were performed to compare conditions with and without *MUS81*. Student T-test is \* P < 0.05, \*\* P < 0.01, and n.s. = not significant. (*C*) Average relative cell viability was calculated at the generation with the lowest average cell density (senescence crisis) for the strains in (*B*). Haploid strains *rad51Δ*, *mus81Δ rad51Δ*, *est2Δ rad51Δ*, and *est2Δ rad51Δ mus81Δ* strains were derived from diploid WDHY3358 as described in Materials and Methods. Remaining haploid strains, *est2Δ rad59Δ, and est2Δ rad59Δ mus81Δ*, were derived from WDHY3366. *WT* and *est2Δ* strains were derived from sporulation of both diploids. (*D*) Terminal telomeres and subtelomeric Y'-elements were freed from the chromosome by *Xho*I digest which cuts internal to the Y'-element region. Depending on the telomere length and subtelomere composition, Southern blot analysis using the Y' probe can monitor both telomere repeat length and the status of the Y'-elements [85]. (*E*) Southern analysis of DNA from early- and post-senescence cells from wild type (*WT*), *est2Δ*, and *est2Δ mus81Δ* genetic backgrounds collected from liquid media. Genomic DNA was probed using an oligonucleotide complementary to the Y'-element region adjacent to the telomere indicated in (*D*). Brackets indicate the terminal *Xho*I fragments and Y'-containing telomeres. Independent survivor isolates are numbered above each lane for both early- and post-senescent cell populations. Plasmid controls (P) are shown in the last two lanes to control for enzymatic digest, with and without *Xho*I (+ and–respectively) and labeled with supercoiled ($P_S$) or linear digested ($P_L$), products. All haploid strains were generated by sporulation of WDHY2961.
(TIF)

**S4 Fig. Mus81 does not exhibit enrichment at telomeres in Type I or Type II survivors.** (*A*) Cells were collected after survival from replicative senescence and subjected to ChIP using anti-Myc antibody [87]. Serial streaks were prepared of *est2Δ rad51Δ* yeast cells with either *MUS81-9MYC* or *mus81-D414,D415A-9MYC* to acquire the survivor strains. Average fold-enrichment of three replicates and a single standard error are presented for each strain. Samples were normalized to input samples and fold-enrichments calculated as Y'subtelomeric DNA over *SAM1* non-telomeric DNA. (*B*) Similar as in (*A*), serial streaks were prepared of *est2Δ rad59Δ* yeast cells with either *MUS81-9MYC* or *mus81-D414,D415A-9MYC* to acquire the survivor strains. Average fold-enrichment of three replicates and a single standard error are presented for each strain. Samples were normalized to samples without antibody and fold-enrichments calculated as Y'subtelomeric DNA over *SAM1* non-telomeric DNA. (*C*) Association of Mcm4-Myc, Esc4-9Myc, and Pol32-9Myc was performed either in *EST2* or telomerase deficient (*est2Δ*) strain backgrounds. Average fold enrichment of three experimental replicates are presented with a single standard of error. Student T-test is * P < 0.05, ** P < 0.01, and n.s. = not significant. Strains were derived from diploids WDHY2961, WDHY2962, and WDHY3007 for *est2Δ mus81Δ*, *est2Δ MUS81-MYC*, and *est2Δ mus81-dd-MYC* respectively. Similar haploids were derived from diploids WDHY5296 (*est2Δ rad51Δ MUS81-9MYC* and *est2Δ rad51Δ mus81-D414,D415A-9MYC*) and WDHY5297 (*est2Δ rad59Δ MUS81-9MYC* and *est2Δ rad59Δ mus81-D414,D415A-9MYC*).
(TIF)

**S5 Fig. Genetic interactions between *RRM3* and *MUS81*.** (*A*) Diploid cells heterozygous for mutations in *MUS81* and *RRM3* were created by mating WDHY3638 and WDHY2272 or WDHY2835. Sporulated haploid spores were allowed to grow on nutrient-rich media for 2–3 days. Colony size was documented and four-spore tetrads were assessed for growth markers related to *rrm3*::KanMX and *mus81*::URA3, while PCR amplification and digest identified *mus81-D414,415A (mus81-dd)*. [Yellow squares = *rrm3Δ*; yellow diamond = *mus81Δ*; yellow circles = *rrm3Δ mus81Δ;* Yellow hexagon (point vertical) = *mus81-dd*; Yellow hexagon (point horizontal) = *rrm3Δ mus81-dd*] (*B*) Indicated strains were assessed by chronic exposure to methyl methanesulfonate (MMS) and hydroxyurea (HU), wild type (W303-RAD5 MATα), *mus81Δ* (WDHY1858), *rrm3Δ* (WDHY3638), and *mus81Δ rrm3Δ* (WDHY3606). (*C*) Viability of haploid yeast in (*B*) was assessed under conditions of chronic exposure to increasing concentrations of MMS or left untreated. Average relative viability was plotted at every concentration for at least 5 clones with a single standard error. (*D*) Schematic representation of a chromosome arm containing a tandem arrangement of two Y' subtelomeric elements and the associated replication intermediates profile by 2D-gel. The Y' elements contain an ARS (black circle) and an *Eco*RI site "E". The presence of telomeric TG tracts, at both the telomere and between the two Y' elements, is shown in red. Digestion with *Eco*RI will produce two fragments: the terminal L1 fragment (blue) and an internal L2 fragment (green) which can be 5.5 or 6.8 kb-long. Both the L1 and L2 fragments mainly migrate as Y structures (y1 and y2). Arrows indicate the expected structures upon replication fork pausing at TG tracts in the *rrm3Δ* mutant: a bubble structure (b1) for the terminal fragment when the distal fork is block at the telomere, and a local increase of signal along the y2 arcs upon stalling at the internal TG tract (67). (*E*) Representative 2D-gel analysis of sub-telomeric and telomeric replication intermediates in asynchronous WT (W303-RAD5), *mus81Δ* (WDHY5102), *rrm3Δ* (WDHY3638) and *mus81Δ rrm3Δ* (WDHY3605) strains. (*F*) Quantification of the replication pausing at the internal TG tract between the two Y' elements. The background-corrected signal was normalized on the L1 spot. Averages and a single standard of error were plotted. n = 2 for WT and

*mus81Δ*, n = 3 for *rrm3Δ*, and n = 4 for *mus81Δ rrm3Δ*.
(TIF)

**S6 Fig. Loss of *MUS81* in *tlc1Δ* cells results in accelerated senescence irrespective of *RAD5* mutation.** (*A*) Serial dilution assays monitoring cell density after 24 hours from an initial inoculate of $5x10^5$ cells. Average cell density is plotted at the given generations for wild type (WT) *RAD5* (n = 4), *mus81Δ RAD5* (n = 3), *tlc1Δ RAD5* (n = 8), *tlc1Δ mus81Δ RAD5* (n = 4), *tlc1Δ mus81Δ rad5-535* (n = 4) strains. Haploid strains were generated by sporulation of WDHY3651 as described in Materials and Methods.
(TIF)

**S7 Fig. Although involved in replication, mutations in RAD5 do not affect cell density in a serial dilution assay.** (*A*) Serial dilution assays monitoring cell density after 24 hours from an initial inoculate of $5x10^5$ cells. Average cell density of at least eight haploid spores is plotted at the given generations for *rad5-535 mus81Δ* (n = 8), *est2Δ* (n = 40), *est2Δ mus81Δ* (n = 26), *est2Δ rad5-535* (n = 8) and *est2Δ mus81Δ rad5-535* (n = 8) strains. Haploid strains in (*A*) were generated by sporulation of WDHY5327 as described in Materials and Methods. (*B*) Haploid yeast strains were assessed by chronic exposure to methyl methanesulfonate (MMS). Strains included *wild type* (W303-RAD5), *rad5-535* (W303), *mus81Δ* (WDHY1858), *mus81Δ rad5-535* (JMY380), *yen1Δ* (WDHY2755), *yen1Δ rad5-535* (WDHY3105), *rad1Δ* (WDHY3106), *rad1Δ rad5-535* (WDHY3161), *slx1Δ* (WDHY3148), and *slx1Δ rad5-535* (WDHY3113).
(TIF)

**S8 Fig. Colony counts after 2- or 5-days incubation.** As part of the serial dilution assay, cell bodies were counted, and predetermined number of cells plated to assess viability. Visible colony forming units were counted regardless of colony size. Average numbers of colonies are presented with one standard error. Haploid strains were generated by sporulation of WDHY3007 (WT, *est2Δ* and *est2Δ mus81Δ)* as described in Materials and Methods.
(TIF)

**S1 Table. *Saccharomyces cerevisiae* strains.**
(DOCX)

**S1 Data. Data file corresponding to Figs 1; 2; 3B and 3C; 6A–6C.** Each strain corresponds to a different data sheet. Identification of lowest cell concentration, viability and statistics data are also included on separate sheets.
(XLSX)

**S2 Data. Data file corresponding to Figs 4B and 5B; S1C Fig; S4A–S4C Fig; S5C Fig; and S5F Fig.**
(XLSX)

**S3 Data. Data file corresponding to S6 Fig.**
(XLSX)

**S4 Data. Data file corresponding to S8 Fig.**
(XLSX)

## Acknowledgments

We thank Adam Bailis, Rodney Rothstein, Steven Elledge for providing strains and Dan Gottschling for providing the plasmids for telomere probe production. We also thank Lifeng Xu for sharing her expertise in telomere biology, and Rinti Mukherjee and Shannon Owens

for comments on the manuscript. Neil Hunter graciously allowed us access to his light microscope.

## Author Contributions

**Conceptualization:** Erin K. Schwartz, Wolf-Dietrich Heyer.

**Formal analysis:** Erin K. Schwartz, Wolf-Dietrich Heyer.

**Funding acquisition:** Wolf-Dietrich Heyer.

**Investigation:** Erin K. Schwartz, Shih-Hsun Hung, Damon Meyer, Aurèle Piazza, Kevin Yan, Becky Xu Hua Fu.

**Project administration:** Wolf-Dietrich Heyer.

**Supervision:** Erin K. Schwartz, Wolf-Dietrich Heyer.

**Writing – original draft:** Erin K. Schwartz.

**Writing – review & editing:** Erin K. Schwartz, Wolf-Dietrich Heyer.

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
