## [Decision Letter · Decision Letter 0]

29 Jan 2020

Dear Wolf,

Thank you very much for submitting your Research Article entitled 'Mus81-Mms4 endonuclease is required for telomere maintenance under replication stress in Saccharomyces cerevisiae' to PLOS Genetics. Your manuscript was fully evaluated at the editorial level and by three independent peer reviewers. The reviewers appreciated the attention to an important topic but identified aspects of the manuscript that need to be addressed.  Much of this revolves around the language used (especially the title) and the possibility of alternative interpretations.  That Mus81 is not required for Type I or Type II survivors seems clear, which was the main point you were trying to may, but exactly why it is needed remains speculative.

We, therefore, ask you to modify the manuscript according to the review recommendations before we can consider your manuscript for acceptance. Your revisions should address the specific points made by each reviewer.

[LINK]

Yours sincerely,

Sue Jinks-Robertson, Ph.D.

Associate Editor

PLOS Genetics

Gregory P. Copenhaver

Editor-in-Chief

PLOS Genetics

Reviewer's Responses to Questions

**Comments to the Authors:**

Reviewer #1: Schwartz et al present evidence that Mus81-Mms4 plays a role in promoting cell viability during telomere-induced senescence. They also show that Mus81 is not required for Type I and II survivors to form. Based on their findings, Schwartz et al propose that loss of telomerase causes replication stress at telomeres, which Mus81-Mms4 is needed to deal with. This is certainly a plausible idea that could account for their data. However, I think that their body of evidence falls a bit short of what would be required to definitively support the bold claim that the title of their paper makes – “Mus81-Mms4 endonuclease is required for telomere maintenance under replication stress in Saccharomyces cerevisiae”. I think they would have a much stronger case if they showed that Mus81 was exclusively enriched at telomeres in telomerase defective cells (e.g. by a calibrated ChIP-seq experiment). As it stands, one could propose that profound changes in gene expression, which are known to occur during cellular senescence, could lead to genome-wide problems that may require Mus81’s attention.

Other comments:

1) Figure 2: The data appear to show that Mus81 is required for the viability of Type I/II survivors (i.e. from ~65 generations onwards the viability of mus81∆ est2∆ cells is less than est2∆ cells). The authors should make a comment about this. The authors should also include data for the relative viability of a mus81∆ single mutant to confirm that it doesn’t differ from wild-type by the same order of magnitude as a mus81∆ est2∆ double mutant differs from an est2∆ single mutant.

2) Figure 2: The data in panels A and B don’t appear to be consistent – in A the lowest relative viability for the est2∆ mutant is ~0.26 – 0.27, whereas in B it is shown to be 0.3.

3) I think that the authors should do more to establish why their data differs so markedly from previous studies. At the very least, they should establish whether this is due to a difference between est2∆ and tlc1∆.

4) There are a few spelling/grammar/typographical errors, e.g. page 19 “platting” and supplemental page 11 “Average fold-enrichment of with three replicates..”

Reviewer #2: Previous studies in human cells suggested a role for MUS81 in extension of telomeres by a recombination-based mechanism, referred to as ALT. In this study, the authors use budding yeast to investigate whether the function of Mus81 in the generation of survivors from telomerase-negative cells is conserved, and to assess whether Mus81 has a more general role in telomere biology.

The authors find that survivors arise from est2 mus81 cells with similar kinetics to est2 cells. Moreover, by use of rad51 and rad59 derivatives of est2 mus81 cells, the authors show that Mus81 is dispensable for the generation of type I or type II survivors. Consistent with these observations, Southern blot analysis of survivors from est2 mus81 showed the typical type II pattern with long, heterogeneous telomeres.

In contrast to previous studies (Azam et al, 2006; Fallet et al, 2014), the authors report accelerated senescence and reduced viability of est2 mus81 cells as compared to the est2 single mutant, which they attribute to a role for Mus81 in resolving replication stress at telomeres. In support of this idea, the authors show that mus81 enhances the MMS and HU sensitivity of est2 and cdc13-1 mutants, and Mus81 in enriched at telomeric DNA in pre-senescent cells. The authors also report additive defects of mus81 and rrm3 in the est2 or cdc13-1 background.

These observations raise the question of whether the requirement for MUS81 in ALT is related to recombination or to the increased replication stress in ALT cells. Thus, the work will be of interest to investigators studying ALT, as well as the structure-selective endonuclease field.

Comments:

I am not completely convinced that cdc13-1 mus81 null cells have reduced viability compared with cdc13-1. Were cell division times calculated? Also, because of the decreased viability of the cdc13-1 mus81 rrm3 triple mutant at 27 C in the absence of MMS, I don’t think the authors can conclude that the triple mutant is more sensitive than the doubles to MMS. They would need to do quantitative survival assays to make this claim.

Do mus81 null rrm3 spores have smaller colony sizes? Some figures show the mus81-dd allele and others the null. In the case of synthetic interaction with rrm3 the authors should be consistent with which allele they use in different assays.

p. 9: The term “replication timing” is normally used to describe the temporal order of replication firing during S phase. Do the authors mean generation time?

p. 11, first line: Use Type I or Type II for consistency.

p. 13. Fig S4 only shows inviability of cdc13-1 cells at 30 C, not G2-M arrest. Fig S4 could be removed. Numerous previous studies have documented the cdc13-1 growth defect at 30 C.

p. 22, line 11: non-permissive should be semi-permissive.

Reviewer #3: Schwartz et al. report that the structure-specific Mus81-Mms4 plays a role in the survival of yeast cells lacking telomerase, but not in either the Type I or Type II forms of repair. Hence the title strikes me as over-stated. I think most readers will interpret the claim that Mus81 is “required for telomere maintenance under replication stress” to mean that it is involved in telomere maintenance in the absence of telomerase or possibly even in the absence of telomere capping. Rather, the results seem to show that Mus81-Mms4 prevents accelerated senescence. To me that’s not the same thing and it makes the point that – despite the starting point of the authors, that Mus81 plays a role in mammalian ALT cells – Mus81 does not obviously play a result in yeast’s two “ALT” pathways. So my suggested title is” Mus81-Mms4 prevents accelerated senescence in telomerase-deficient cells.”

In new data, the authors show some sort of synergy between agents that cause replication stalling (MMS and HU) and telomere dysfunction, as both est2 cells (well before senescence) and cdc13-1 cells show increased drug sensitivity in the absence of Mus81. Exactly what is being measured here is not clear. Both telomere damage and MMS/HU trigger a DNA damage response. It seems to me quite likely that the role of Mus81 in all of these assays is checkpoint-related and that combinations that increase the damage signal may impair the ability of haploid cells to “adapt” and to get to the point where survival through one of the repair pathways is possible. For example, Toczyski showed that cdc13-1 cells fail to resume cell division in mutants such as cdc5-ad or ckb1. Longhese showed similarly that overexpressing Ddc2, a key component of the DNA damage checkpoint, also prevented cells with a single chromosome break from resuming cell cycle progression. Since est2-deleted cells also elevate the checkpoint, perhaps the real problem here has nothing to do with recombination per se but with getting to the point where recombination can take place.

1. Hence it might be informative to repeat the assay done by Toczyski, using est2 and cdc13-1 in combination with cdc15-ts as a way to distinguish those cells that completed mitosis and those arrested at the G2/M boundary.

2. The fact that mus81 est2 cells lose viability more rapidly can have two interpretations. First, there may indeed be some sort of accelerated inviability associated with replication problems near telomeres. Alternatively, there could simply be a delay in the formation of the first viable survivors.

Does viability show the same decrease in a rad52 mutant, where there are no survivors and the drop in viable cells will measure the rate of a fatal telomere shortening?

By looking at telomere length directly, is there an obvious change in the distribution of telomere lengths at or before the time of “crisis”? Fig. S3 looks at telomere length but since the gels are separate, I don’t think it’s possible to evaluate from these images whether there are more short telomeres in est2 mus81 than in est2 alone.

3. Mus81 is recruited to senescent telomeres. I agree it probably isn’t just a “passenger” but it is worth noting that Slx4 anchors a number of different nucleases and repair proteins and if it is recruited, its partners may come for the ride. Even inactive partners that still associate with the scaffold.

**Have all data underlying the figures and results presented in the manuscript been provided?**

Reviewer #1: Yes

Reviewer #2: Yes

Reviewer #3: Yes

PLOS authors have the option to publish the peer review history of their article (what does this mean?). If published, this will include your full peer review and any attached files.

Reviewer #1: No

Reviewer #2: No

Reviewer #3: No

---

## [Decision Letter · Decision Letter 1]

27 Apr 2020

Dear Wolf,

Thank you very much for submitting a revision of your Research Article entitled 'Saccharomyces cerevisiae Mus81-Mms4 prevents accelerated senescence in telomerase-deficient cells' to PLOS Genetics. Your manuscript was fully evaluated at the editorial level and by one of the initial peer reviewers. Given the current COVID-19 situation, the reviewer requested only a few additional textual changes that I think are reasonable.  Once these are addressed, the manuscript will be accepted without further review.

[LINK]

Yours sincerely,

Sue Jinks-Robertson, Ph.D.

Associate Editor

PLOS Genetics

Gregory P. Copenhaver

Editor-in-Chief

PLOS Genetics

Reviewer's Responses to Questions

**Comments to the Authors:**

Reviewer #3: attachment

**Have all data underlying the figures and results presented in the manuscript been provided?**

Reviewer #3: Yes

PLOS authors have the option to publish the peer review history of their article (what does this mean?). If published, this will include your full peer review and any attached files.

Reviewer #3: No

---

## [Editor Report · Decision Letter 2]

30 Apr 2020

Dear Wolf,

We are pleased to inform you that your manuscript entitled "Saccharomyces cerevisiae Mus81-Mms4 prevents accelerated senescence in telomerase-deficient cells" has been editorially accepted for publication in PLOS Genetics. Congratulations!

Yours sincerely,

Sue Jinks-Robertson, Ph.D.

Associate Editor

PLOS Genetics

Gregory P. Copenhaver

Editor-in-Chief

PLOS Genetics

**Data Deposition**

http://datadryad.org/submit?journalID=pgenetics&manu=PGENETICS-D-19-02131R2

**Press Queries**

---

## [Editor Report · Acceptance letter]

21 May 2020

PGENETICS-D-19-02131R2 

Saccharomyces cerevisiae Mus81-Mms4 prevents accelerated senescence in telomerase-deficient cells 

Dear Dr Heyer, 

We are pleased to inform you that your manuscript entitled "Saccharomyces cerevisiae Mus81-Mms4 prevents accelerated senescence in telomerase-deficient cells" has been formally accepted for publication in PLOS Genetics! Your manuscript is now with our production department and you will be notified of the publication date in due course.

With kind regards,

Jason Norris

PLOS Genetics

On behalf of:
